# Metal 3D nanoprinting with coupled fields

Bingyan Liu[1], Shirong Liu[1], Vasanthan Devaraj [2], Yuxiang Yin[1], Yueqi Zhang[1], Jingui Ai[1], Yaochen Han[1] & Jicheng Feng [1] ✉

Metallized arrays of three-dimensional (3D) nanoarchitectures offer new and exciting prospects in nanophotonics and nanoelectronics. Engineering these repeating nanoarchitectures, which have dimensions smaller than the wavelength of the light source, enables in-depth investigation of unprecedented light–matter interactions. Conventional metal nanomanufacturing relies largely on lithographic methods that are limited regarding the choice of materials and machine write time and are restricted to flat patterns and rigid structures. Herein, we present a 3D nanoprinter devised to fabricate flexible arrays of 3D metallic nanoarchitectures over areas up to $4 \times 4$ mm$^2$ within 20 min. By suitably adjusting the electric and flow fields, metal lines as narrow as 14 nm were printed. We also demonstrate the key ability to print a wide variety of materials ranging from single metals, alloys to multimaterials. In addition, the optical properties of the as-printed 3D nanoarchitectures can be tailored by varying the material, geometry, feature size, and periodic arrangement. The custom-designed and custom-built 3D nanoprinter not only combines metal 3D printing with nanoscale precision but also decouples the materials from the printing process, thereby yielding opportunities to advance future nanophotonics and semiconductor devices.

Optical metamaterials[1,2] and nanoplasmonics[3,4] have effectuated a paradigm shift in conventional optics with the development of nanotechnology[5]. Nanoplasmonics of metal nanostructures displays exciting and technologically beneficial capabilities that arise from the coupling of incident light to the collective motion of the conduction electrons[3]. The excitation, propagation, and localization of these plasmons can be tailored by altering the materials, geometries, and sizes of metal nanostructures[6]. Engineering the arrangement of repeating structures that have dimensions smaller than the wavelength of the light source enables in-depth investigation of unprecedented light–matter interaction phenomena[7]. Microelectronics necessitates the fabrication of three-dimensional (3D) metallic materials that occupy $z$-axis space while maintaining the miniaturized areal dimension, which remains challenging, especially at small scales[8–10]. In contrast to synthesis techniques for lithographic and two-dimensional (2D) materials, the capability to control the height of nanostructures in the $z$-axis direction can be achieved via 3D nanoprinting techniques. Structural flexibility fulfills the requirements of advanced packaging, and the freedom this affords has given rise to new developments in

high-density and high-aspect-ratio microbumps, fine interconnects, and transistors[11–16].

Modern tools, such as lasers[17], electron/ion beams[18], or micropipettes[19,20], have been developed for the fabrication of 3D nanostructures. However, these tools are generally restricted in terms of the materials they can process, their downsizing ability, and their efficiency with respect to printing periodic arrays of nanostructures. Micro-stereolithography is a technique that is commonly used for printing small features; this technique utilizes polymeric templates for subsequent metallization[21–24]. However, the resulting (composite) materials rarely retain their metallic characteristics without stringent post-deposition treatments. In a recent demonstration[24], a laser source was used to shrink a hydrogel framework for printing multimaterials, with purity being the primary concern. The development of 3D nanoprinting techniques based on aerosols has advanced considerably in different respects[25–28]. Nevertheless, the ability to print multimaterials has remained elusive, mainly owing to its strong dependence on materials physics[29,30]. Although a single pillar consisting of four different

[1]School of Physical Science and Technology, ShanghaiTech University, Shanghai, China. [2]Bio-IT Fusion Technology Research Institute, Pusan National University, Busan, Republic of Korea. ✉e-mail: fengjch@shanghaitech.edu.cn

metals (Au, Ag, Cu, and Pd) was successfully printed, the dimensions of these metal layers varied from one another[25], which is likely related to the difficulties in decoupling materials for printing. Moreover, these studies also demonstrated nanostructures printed within a sub-1-mm² area, with the smallest feature size reported as 70–100 nm[25–28]. Structural uniformity and miniaturization were often compromised while printing structures spread across larger areas[25,27,30]. In addition to complex architectures, small feature sizes, and high printing speeds, the ability to print multimaterials is a crucial aspect that must be considered while printing nanostructures. In metal 3D nanoprinting, the properties of the material play a decisive role in end-product applications in fields such as nanoelectronics[31], nanophotonics[32–34], microfluidics[35], and microelectromechanical systems[36]. In this regard, scalable techniques for metal 3D nanoprinting are a prerequisite to effectively fabricate structures comprising multimaterials with desirable architectures.

Herein, we present a 3D nanoprinter devised to print structures consisting of multiple materials and including various periodic arrays and complex nanoarchitectures, as well as the investigation of their optical properties via several methods. By controlling the applied electric field and adjusting the flow field, we could precisely select the size of nanoparticles (NPs) as building blocks for in situ printing, despite their material differences (Pt, Au, Ag, Pd, and Ni–Ti, Au–Ag, Ni–Cr–Co–Mo–Ag alloys). In practice, the use of two flows creates a double-layer system (Fig. 1a), comprising a top

layer with the NPs (i.e., an aerosol flow) and a bottom layer without the NPs (i.e., a high-purity inert gas). This flow pattern enabled us to print lines as narrow as 14 nm. Considering that uniform fields can be maintained with the aid of the double-layer flow, the spatial distribution of the electric field was designed such that it was reflected in the geometries of the printed nanostructures. Hereafter, this spatial distribution is referred to as "field maps." A variety of 3D nanostructures consisting of multiple materials (ranging from single metals to multimaterials) with different geometries were printed to cover areas as large as 4 × 4 mm², thereby demonstrating that the materials could be successfully decoupled from the printing process. The characteristic size of the local field was consistent with the predictions based on our model, confirming that the proposed 3D nanoprinter has the ability to exercise precise control. Using the library of 3D nanostructures, we studied their optical properties, including their interactions with a light source (wavelengths ranging from visible to infrared (IR)). The electron excitations of the 3D-printed nanostructures were characterized and analyzed by acquiring their electron energy-loss spectra with the aid of Auger electron microscopy. The homemade 3D nanoprinter enables the possibilities for investigating and engineering nanoplasmonics with 3D nanostructures, offering a research tool for manipulating light–matter interactions at the nanoscale. Extending the concept to systems comprising other materials is expected to lead to breakthroughs in various applications in the fields of nanoelectronics, nanophotonics, and sensors[32–34].

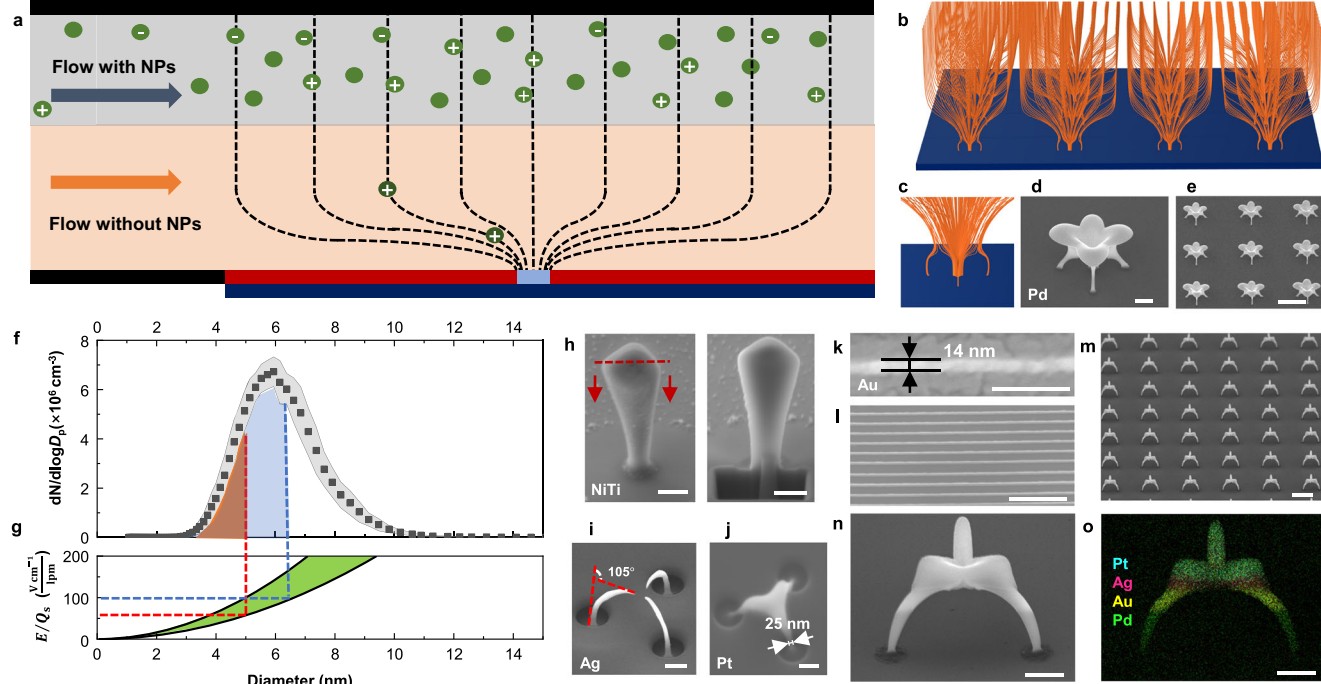

**Fig. 1 | Coupled electric and flow fields in 3D nanoprinting. a** Schematic of the custom-built 3D nanoprinter with a clean-gas layer that is parallel to an aerosol flow. Dashed curves represent the electric-field lines, and the green dots denote charged and neutral NPs. To ensure the uniformity of the printed nanostructures, the clean-gas layer must be sandwiched between the aerosol flow and substrate surface to prevent NP diffusion to the printed area and to remove the NP cloud[27]. **b** Simulation results for the 3D maps of electric fields in periodic patterns and an enlarged view (**c**). **d**, **e** SEM images of a 3D-printed nanoarchitecture (scale bar: 1 μm) and the corresponding array (scale bar: 5 μm) defined by field maps. **f** Particle-size distribution of Au NPs with the gray shadow indicating five repeated measurements. The area under the lognormal curve represents the total concentration of NPs, each of which carries a positive elementary charge. The orange and the light-blue areas underneath the curve represent the two different total

concentrations of the NPs after size selection according to (**g**). **g** Particle size ($D_p$) as a function of the ratio between the electric-field strength and flow rate of the clean gas (lpm: liter per minute). The light-green region indicates the variation in the geometric volume of the nanoprinter (from the lower to upper edge, the volume decreases by ~53%). **h** SEM images of a structure before and after FIB processing (scale bar: 500 nm), showing that the interior is densely packed. SEM images for high-aspect-ratio (=15) bending nanowires (**i**) (diameter: 200 nm; length: 3000 nm; bending angle: 105°). **j**–**l** Demonstrated nanoscale precision. A Pt nanostructure with a minimum feature size of 25 nm (**j**) and 14-nm-wide Au lines (**k**, **l**). The scale bars are 100 nm for (**j**, **k**), 1 μm for (**i**, **l**, **n**, **o**), and 5 μm for (**m**). **m**, **n** Multimaterial (Pt, Ag, Au, and Pd) printing, as confirmed via EDS mapping in **o**. The tilt angles used are presented in Supplementary Table 1. Source data are provided as a Source Data file.

## Results

### In situ printing of size-selected NPs by coupling the electric and flow fields

The devised 3D nanoprinter comprises a particle source and printing zone (Supplementary Fig. 1). This printing was performed under ambient conditions, and it required only gas-phase methods for synthesizing aerosol NPs. Some of these particles were charged and served as the building blocks for printing by mapping the electric field to 3D geometries with the help of a patterned photoresist (PR). The NPs have a geometric mean diameter of 3–5 nm, with a geometric standard deviation of 1.2–1.4 (Supplementary Fig. 2). In conventional nanoprinters, the aerosol NPs flow toward the printing area[27], resulting in a limited region with uniform structures and the inability to achieve multimaterial printing. To solve these problems, we introduced a double-layer flow (Fig. 1a) in our crafted 3D nanoprinter—the top layer carries NPs in the form of an aerosol, and the bottom layer is constantly flushed by clean gas flowing parallel to the substrate. The layer of clean gas was introduced to provide a sheathing effect that prevented the deposition of neutral NPs in the printed area (Supplementary Fig. 3). This strategy was utilized in tandem with the application of an electric field, allowing NPs to be precisely selected based on differences in their electrical mobilities. We arrived at a solution for selecting the particle size by adjusting the flow rate of the clean gas and the strength of the electric field (details are presented in Supplementary Discussion 1) as follows:

$$\frac{D_p}{C} = \left(S\frac{e}{3\pi\eta}\right)\frac{E}{Q_s} \tag{1}$$

where the Cunningham correction factor ($C$) is a function of the particle size ($D_p$); $S$ represents the scanning surface area, which reflects the geometric volume of the printing chamber; $e$ denotes an elementary charge; $\eta$ indicates the dynamic viscosity; $E$ symbolizes the strength of the external field; $Q_s$ depicts the flow rate of the clean gas.

Our 3D nanoprinter relies on coupling between the electric and flow fields to print structures consisting of charged NPs within a specific size range, despite the differences in their constituents. In Fig. 1a, the dark-red layer comprises the PR, which covers a Si wafer (the substrate). The light-blue area in the center contains an array of openings that were fabricated using lithographic techniques, and the holey arrays can be designed to take a variety of forms (Supplementary Table 2); a representative form is shown as an enlarged view with 3D map of the fields (cf. Fig. 1b, c). These fields determine the architectures of the printed structures (Fig. 1d, e), whose locations were dictated by the holes. The aerosol flow (i.e., the gas flow carrying the NPs) contains a fraction of gas ions, which are subsequently deposited onto the PR layer to form a repelling field, which only has a short-range influence. Thus, the external field is reconfigured into the shape portrayed by the dashed curves. When the charged NPs enter the electric field, some of them are printed, and those with the opposite polarity are directed upwards. By controlling the strength of the external field and flow rate of the clean gas, the overall concentration of NPs can be selected, along with a specific size range, and these NPs are represented by the area under the lognormal size-distribution curve (i.e., the brown and blue areas in Fig. 1f). For example, the red line in Fig. 1g corresponds to the brown area representing the selection of NPs with a size < 5 nm and a total concentration of $4 \times 10^5 \, \text{cm}^{-3}$ for printing. To maintain the guideline feature of Eq. (1), we carefully controlled the density of the NPs to prevent particle agglomeration[37] (thereby decreasing the likelihood of the deposited NPs carrying multiple charges that are ill-defined) and reduce the space charge effect[38] (density of charged NPs < $1 \times 10^8 \, \text{cm}^{-3}$). The same working principle applies to the indicated area and dashed lines in Fig. 1f, g, respectively. The light-green area in Fig. 1g indicates the

volume variation of the 3D nanoprinter; details regarding this aspect are presented in Supplementary Fig. 4. To confirm that the materials were densely packed inside the structure, focused-ion beam (FIB) milling was utilized to obtain the interior of the printed Ni−Ti nanostructure, (Fig. 1h), whose SEM image showed no obvious porosity. Compared to porous structures, their non-porous counterparts reportedly have superior mechanical and electrical properties[39–42]. Because the printing occurred in an inert gas environment, the NP building blocks remained highly pure[30]. The deposition of these pure NPs led to local coalescence to form compact nanostructures that exhibited electrical and mechanical properties comparable to those of the bulk material[25]. Detailed results for the interior structures after FIB treatment are presented in Supplementary Fig. 5. Previously, 2D metasurfaces whose bending angles could be varied to manipulate the phase mutation of the scattered light were developed[43], and we fabricated these metasurfaces in 3D forms (Fig. 1i). This periodic arrangement with different bending angles (Supplementary Fig. 6) and intergaps can be used to control light such that it is propagated in the desired direction[43]. The miniaturization ability of our nanoprinting technique was demonstrated by printing metals with a minimum feature size of 25 nm (Fig. 1j; Supplementary Fig. 7a) and a line width of 14 nm (Fig. 1k, l; Supplementary Fig. 7b). The printed 14-nm wide lines demonstrate the powerful ability of our technique to transcend the minimal lateral dimensions for metal structures and lay the foundation for subsequent printing of complex 3D nanostructures while maintaining the demonstrated resolution. Technically speaking, the metal lines should be more accurately referred to as "nanowalls" (Supplementary Fig. 7e, f), as they had an aspect ratio of ca. 3, with a height, width, and length of 60, 24, and 10,000 nm, respectively. For the sake of comparison and consistency with the literature, we use "line width" as the standard terminology throughout this manuscript. As far as we know, this ability to print features with an approximate width of 10 nm and to achieve flexibility of nanoarchitecturing represents a milestone in metal 3D printing[21,22,44]. As a proof of concept, the results in Fig. 1m, n indicate that the map of the fields was flexibly controlled to print periodic arrays of nanoarchitectures consisting of multimaterials (printed in order of Pd, Au, Ag, and Pt), as confirmed by energy-dispersive X-ray spectroscopy (EDS) mapping (Fig. 1o). Additional results pertaining to printing multimaterials are presented below and in Supplementary Fig. 8 and Supplementary Table 3.

The ability to select the size of NPs is an important development, because this decouples the printing process from the particle source. This key development offers the following three benefits:

1. Any material can be printed, as long as NP size selection is performed.
2. Size selection yields a narrow size range to improve the compactness of the printed nanostructures
3. Features that are ~10 nm wide can be obtained by pushing the brown region in Fig. 1f leftward.

Furthermore, we demonstrated the scalability of our devised 3D nanoprinter in that it only required 20 min to efficiently print $8000 \times 8000$ uniform 3D nanostructures over an area of $4 \times 4 \, \text{mm}^2$ in one pass. Printing across a large area was achieved by applying a pulsed electric field to the printing zone, because charged NPs in an aerosol can be distributed over the entire area designated for printing the nanostructures (Supplementary Fig. 9 and Supplementary Discussion 2). Thus, the proposed nanoprinter provides a minimalist and cost-effective way to increase 3D printing productivity, extending its application beyond prototyping.

### Mapping local fields via printed nanostructures

In addition to the central role of the double-layer flow system, we investigated the map of the local fields with the printed nanostructures for a deeper understanding of the field-defined geometries. The domain of the ions deposited on the PR surface formed a local field

around the openings. This local field can be imagined as a cushion that competes with the externally applied constant field to take a semi-circular shape (Fig. 2a). Two adjacent half-circles squeeze the external-field lines into an electric funnel[27] that forms the guiding path for the charged NPs (Fig. 2a). Here we wished to quantify the correlations between the two fields.

Initially, according to the analytical model, we defined the cushion height ($h_c$) and characteristic distance ($R$), as schematically illustrated in Fig. 2a, along with other geometric parameters (the PR thickness ($d$) and hole size ($W$)). For a simpler representation, these parameters are reflected by the arrangement shown in the SEM image presented in Fig. 2b. The cushion heights for six different $R$ values (corresponding to six rows of the printed nanostructures in Fig. 2e–i) are presented in Supplementary Fig. 10. The clean gas continuously flushes the printed surface, thereby eliminating the influence of the Debye length on cloud deposition[27]. In accordance with the conservation of field lines, the outermost field line of the electric funnel eventually touches the PR surface[27] for a single hole pattern with a sufficiently large $R$. Considering this scenario, we defined the clean distance $L_c$, expressed as $L_c = \frac{W\sigma}{2\varepsilon_r\varepsilon_0 E}$, where $\sigma$ denotes the charge density; $\varepsilon_0$ and $\varepsilon_r$ symbolize the permittivity of free space and relative gas permittivity, respectively; $E$ represents the strength of the external field. Under our experimental conditions, $L_c$ is ~8 μm, which is greater than the values of $R$ used herein (the maximum $R$ value was 7 μm). Accordingly, we leveraged $R$ to estimate the cushion height as $h_c = \sqrt{\frac{Rd\sigma}{4\pi\varepsilon_r\varepsilon_0 E}}$. When

$R < L_c$, the curve is parabolic. Otherwise, the cushion height levels off and is rendered independent of $R$, as $L_c$ must replace $R$ for estimating $h_c = \sqrt{\frac{Wd\sigma^2}{8\pi\varepsilon_r^2\varepsilon_0^2 E^2}}$. Figure 2c plots this tendency with potentials of 600 and 800 V and the charge density fixed to $\sigma = 5.67 \times 10^{-7}$ C cm$^{-2}$. These scenarios yield two different clean distances: $L_{c1} = 7.8$ and $L_{c2} = 5.9$ μm. Moreover, we compared the predicted values of $h_c$ with the measured results. Changing the potential requires the flow rate to be changed to maintain the size selection (Eq. (1); details are summarized in Supplementary Table 4). A fixed charge density causes the measured values to deviate from those predicted by the model, as shown in Fig. 2d (the measured points are clustered despite the changes in the potential and $R$). This tendency is indicated by the curve that was plotted using data collected by conducting a set of experiments with six different $R$ values at each fixed potential. This deviation is attributed to the self-adjustment of the local field according to the strength of the external field. This led us to conclude that $\sigma/E$ is nearly unchanged, indicating that the cushion height also remains unchanged at different potentials. Figure 2d also suggests that the charge density is independent of the geometric parameters of the hole patterns. The model was developed on the basis of a theoretical framework on electrostatics, and it is independent of the materials. The parameter $h_c$ characterizes the domain governed by the local field and is related to the feature size. As the motion of singly charged NPs is the same in this field configuration, their materials play no role in the printing process. The model prediction is verified by the printed Au–Ag nanostructures,

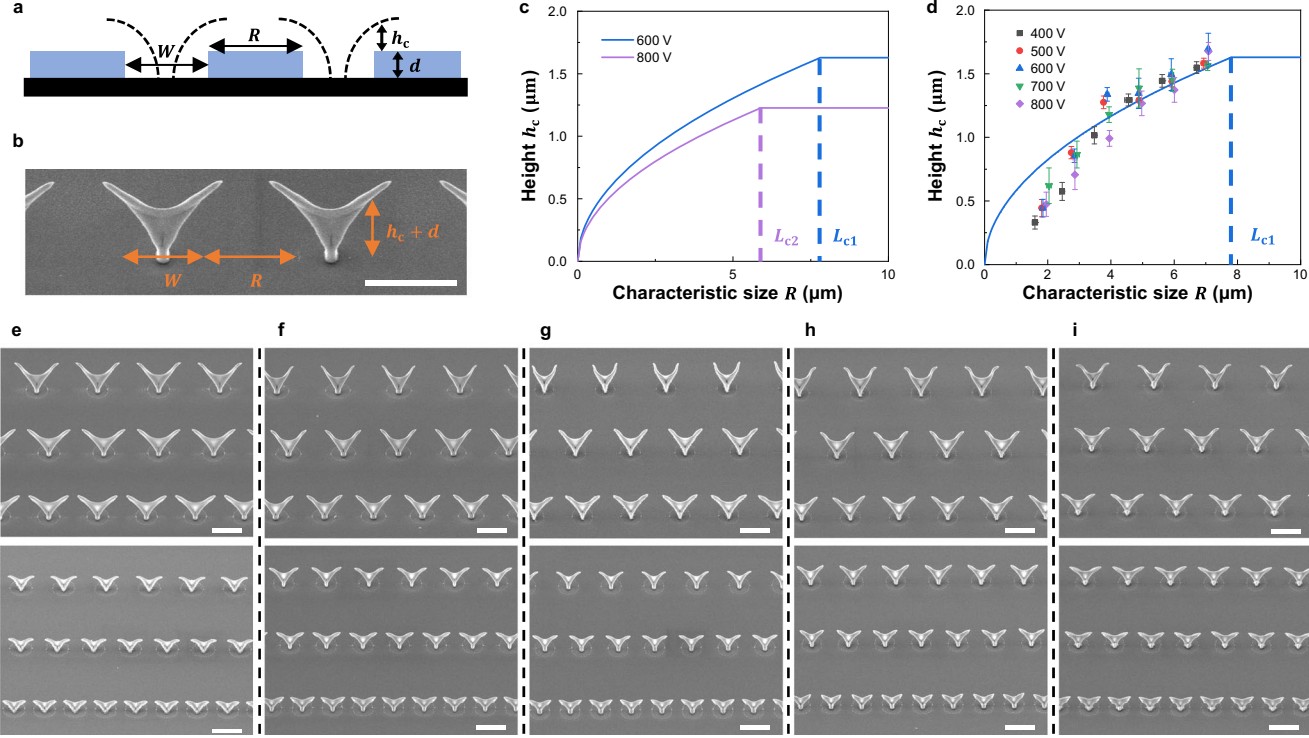

**Fig. 2 | Mapping the fields via 3D-printed nanostructures. a** Schematic of the patterns used to form the local fields, which are known as cushion fields[27]. They can be considered to have a half-spherical shape with a height of $h_c$. The dielectric layer has a thickness of $d$, and the patterned parts with and without the dielectric correspond to characteristic sizes of $R$ and $W$, respectively. **b** All these geometric parameters are mirrored by the SEM image of the printed nanostructures. **c** Predicted relationship between the cushion height ($h_c$) and $R$. The relationship is characterized by a parabolic curve followed by a plateau. The turning point represents the clean distance ($L_c$), which is defined as the point at which the fields at a macroscale and local scale compete with each other. **d** Experimental results confirming the relationship plotted in **c** by varying the strength of the applied

electric field. To maintain the operating parameters for the particle source, we vary the flow rates to maintain a fixed ratio of $E$ to $Q$ (Eq. (1)). The error bars represent the standard deviation, $n = 8$–16 independent replicates (Supplementary Fig. 10). **e–i** SEM images of 3D nanostructures printed at five different potentials (400, 500, 600, 700, and 800 V). The substrate has an array of opening holes with different values of $R$ (2, 3, 4, 5, 6, and 7 μm), which represents the distance between the edges of neighboring holes and the value thereof is fixed for the same row of printed nanostructures. The scale bar represents 5 μm throughout. The tilt angles used for SEM imaging are presented in Supplementary Table 1. Source data are provided as a Source Data file.

but it is also valid for other materials, as different materials are printed to still maintain similar architectures, as shown below.

## Printing multiple materials: periodic arrays of various nanoarchitectures

We demonstrated that the double-layer flow decoupled the materials from the printing process. Accordingly, we successfully mapped the local fields with the printed nanostructures. The exploitation of this advantageous effect also enabled us to print intricate complex architectures consisting of multiple materials (Figs. 3 and 4). The flow rate of the cleaning gas was set from 0.8 to 1.2 liters per minute (lpm) and the strength of the external field ranged from 6250 to 7500 V cm$^{-1}$. Under these conditions, we successfully printed nanostructure arrays comprising the single metals from Au, Ag, Pd, or Pt and the alloys made of Au–Ag, Ni–Ti, or Ni–Cr–Co–Mo–Ag (Fig. 3 and Supplementary Fig. 11). EDS was employed (Figs. 4 and 5 and Supplementary Fig. 12) to verify the compositions of the structures, where the colors yellow, pink, green, and blue correspond to Au, Ag, Pd, and Pt, respectively. The EDS results for a single nanoarchitecture from the arrays in Fig. 3 are correspondingly arranged in Fig. 4. The EDS analysis reveals that the metal(s) content exceeded 94 wt.% (Supplementary Fig. 12), demonstrating a higher material purity than that achieved with other techniques[21,22,44]. The trace amounts of impurities can be attributed to sample transport/storage under ambient conditions. We also managed to print nanoarchitectures comprising multimaterials (Fig. 5a–c, Supplementary Fig. 8). Changing the order in which materials are arranged within a single nanoarchitecture does not give rise to structural variations (Fig. 5a, b). Compared with the printing of a variety of single materials, multimaterial printing maintains the desired nanoarchitectures (Fig. 5c, d). The in situ combination of different aerosol phases readily allows instantaneous switching to a broad range of materials[45], a unique feature for printing multimaterial structures. In our study, aerosols containing NPs of four different metals (Au, Pt, Ag, and Pd) were mixed, and they were then transported to the printing zone to demonstrate a different strategy for fabricating nanostructures consisting of multiple materials (Supplementary Fig. 8). This strategy is achieved via NP–NP mixing, while the multimaterials are also printed in segments with different forms within a single nanoarchitecture (Fig. 5a–c) and in atomic mixing (i.e., alloys indicated in the Au–Ag columns of Figs. 3 and 4 and Supplementary Fig. 8). Figure 5d–h demonstrates the ability to print similar nanoarchitectures despite their different materials (supporting the general applicability of the model developed for estimating the field maps), by exploiting the aforementioned advantage that the material is entirely independent of the printing process. This is because the nanoscale fields guide the singly charged NPs, regardless of their constituent materials. Electronegativity may have an influence on the charge probability of different materials, but this property only determines the charge probability (i.e., the number concentration of charged particles). To compensate for this difference in the amount of material, only the printing time would have to be adjusted accordingly. This important capability opens several opportunities while considering various forms of nanoarchitectures. It is not only advantageous for revealing material-dependent plasmonic behaviors[46] but also useful for other studies[47,48]. This achievement represents an improvement over the state-of-the-art techniques used in the field of microscale 3D printing. Most of these techniques are only capable of handling a few types of metals[44] and have difficulties in printing alloys[30] and maintaining the purity of materials.

## Optical properties of various 3D-printed nanoarchitectures

Thus far, we discussed the ability to print multiple materials with various nanoarchitectures and large arrays with proven uniformity. Hereafter, their interactions are characterized via IR spectroscopy (Supplementary Fig. 13) with a wavelength matching the feature sizes

of the 3D-printed structures presented in Fig. 6a, b. In these periodic arrays of printed nanostructures, each primary unit consists of either three or four "ballet feet"-like structures (Fig. 6c, d). The IR measurements indicate notable differences in the absorption peaks (marked as 1, 2, and 3), as shown in Fig. 6e. These differences are attributed to the resonating effect that results from changing the structures and their arrangement. Notably, their resonance wavelengths lie at $\lambda_r = 8.34$ and 8.65 μm (Fig. 6f), respectively. At these wavelengths, we simulated the electric-field profiles and corresponding mappings of the surface charge density distribution along the XY cross-section. The simulation results convey greater near-field strength for the results in Fig. 6h than for those in Fig. 6g. This result is supported by the differences in the dipolar mode characteristics; in Fig. 6i, j, the yellow circles and arrows indicate the flow directions of the surface charge (simulation details are reported in Supplementary Fig. 14).

Metal surface plasmons are collective excitations of electrons that result in highly confined electromagnetic modes, which can be stimulated via an electron-beam source. To investigate this behavior of the 3D-printed nanostructures, we used Auger electron microscopy to acquire their electron energy-loss spectra. The electron energy-loss spectroscopy (EELS) capability is usually provided by transmission electron microscopes to disperse inelastic electrons from a lamellar sample. Reflection electron energy-loss spectroscopy (REELS)[49] is more suitable for obtaining structure-dependent information as it obviates the need to slice the nanostructures. The REELS results (Fig. 6k) reveal that the signals from the Au 3D nanostructures differed significantly from those of their bulk counterparts[50,51]. This difference is attributed to the unique geometries of our 3D-printed nanostructures. The geometric information is supported by the REELS results depicted in orange and blue (Fig. 6k), which correspond to different locations (orange and blue dots) on the same nanostructure for various nanoarchitectures (the results for the Pd nanostructures are shown in Supplementary Fig. 15). The changes in the REELS peaks are characterized with respect to an array of 3D nanostructures. In this regard, the spectrum in the bottom panel exhibits a peak (1.87 eV) that is distinguishable from those of the other nanostructures (1.43, 1.16, 1.1, and 1.4 eV for the top four nanostructures, obtained from the blue spectra containing structural information). This result is likely associated with miniaturized features. Notably, such REELS differences are tailored by the geometries of our nanoarchitectures. Subsequently, we reduced the feature size to ca. 100 nm and printed the features in periodic arrays. Their reflectivity values were measured using an angle-resolved spectrum system in the micro-region with a visible light source, and the results indicate that the reflectance depends on the angles of the light emission (the reflectivity values in Fig. 6l–o have "V"-shaped configurations). The array of the nanostructures can be changed to tune the reflectivity (Fig. 6l–o). To verify this mechanism, optical images (taken at normal angle) underline these differences in colors for these periodic arrays of nanostructures with pitches of 400, 500, 600, and 700 nm, respectively. The flexibility to control the architectures and materials with our printing technique provides additional freedom for regulating the electromagnetic responses compared to those capable of handling only a few types of materials and geometries. Compelling advantages for adopting this printing technique for fabricating metamaterials/metasurfaces are that the printed 3D nanostructures can be fabricated to be angle-sensitive to different light incidences, and their 3D structural characteristics and materials can also be used to tune the plasmonics[52,53].

## Discussion

In this study, we devised a 3D nanoprinter that utilizes electric and flow fields to print periodic arrays of intricate 3D nanoarchitectures of multiple materials with high uniformity, efficiency, and purity. The electric and flow fields are coupled to enable the sizes of NPs to be selected based on differences in their electrical mobility. Consequently, the influence of the material is eliminated, which is beneficial for

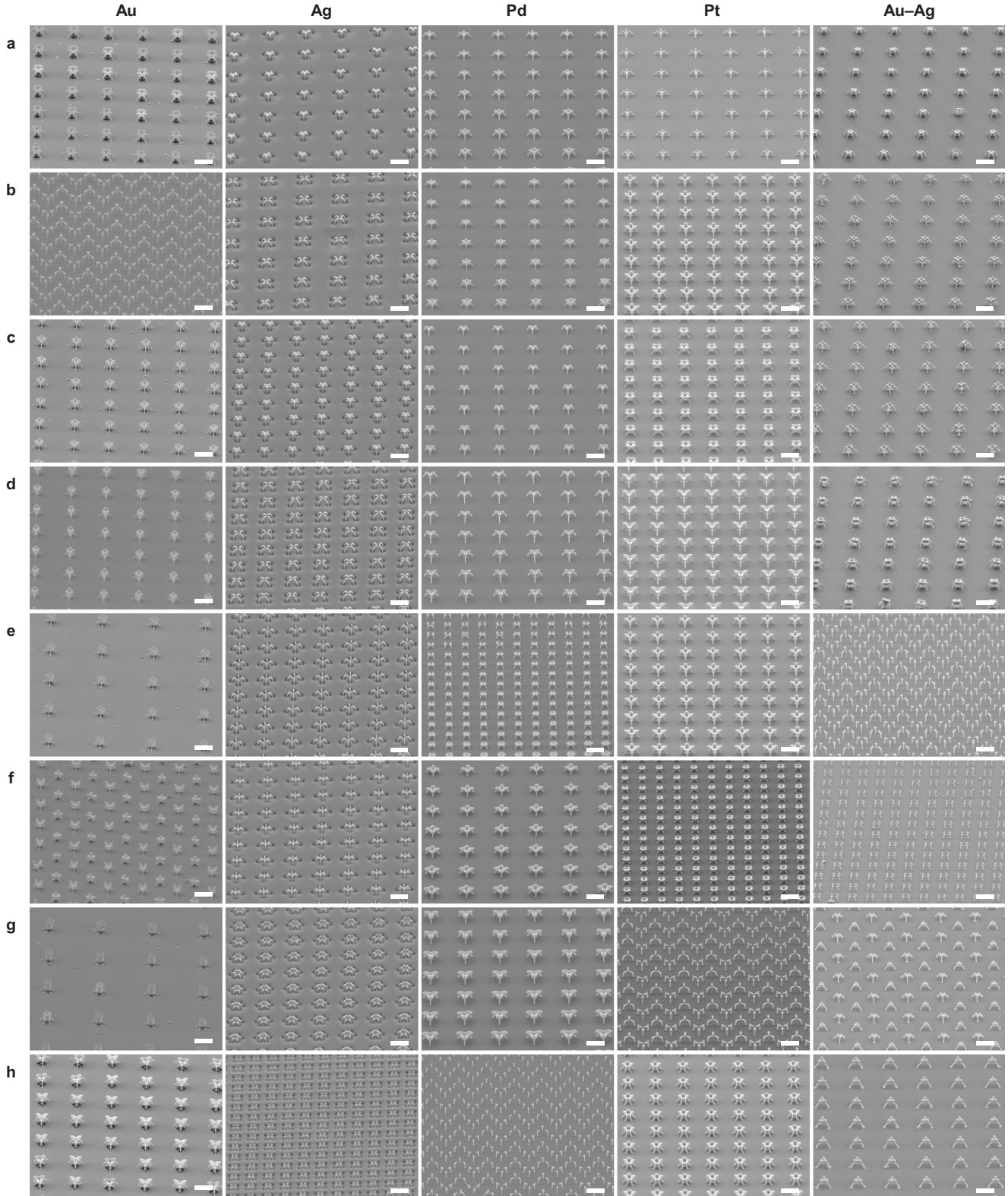

**Fig. 3 | Periodic arrays of 3D-printed metal nanoarchitectures. a–h** SEM images presenting arrays of geometrically different 3D nanostructures comprising Au, Ag, Pd, Pt, and Au–Ag, as indicated above each column, respectively. The 3D nano-printer prints structures with different geometries, regardless of the differences among the materials. Herein, the scale bar represents 5 µm throughout. The enlarged views of a single nanoarchitecture from the corresponding metallic arrays are presented in Fig. 4 to show details of the intricate architectures and materials. The tilt angles for SEM imaging are reported in Supplementary Table 1. Pattern designs for the substrates are presented in Supplementary Table 5.

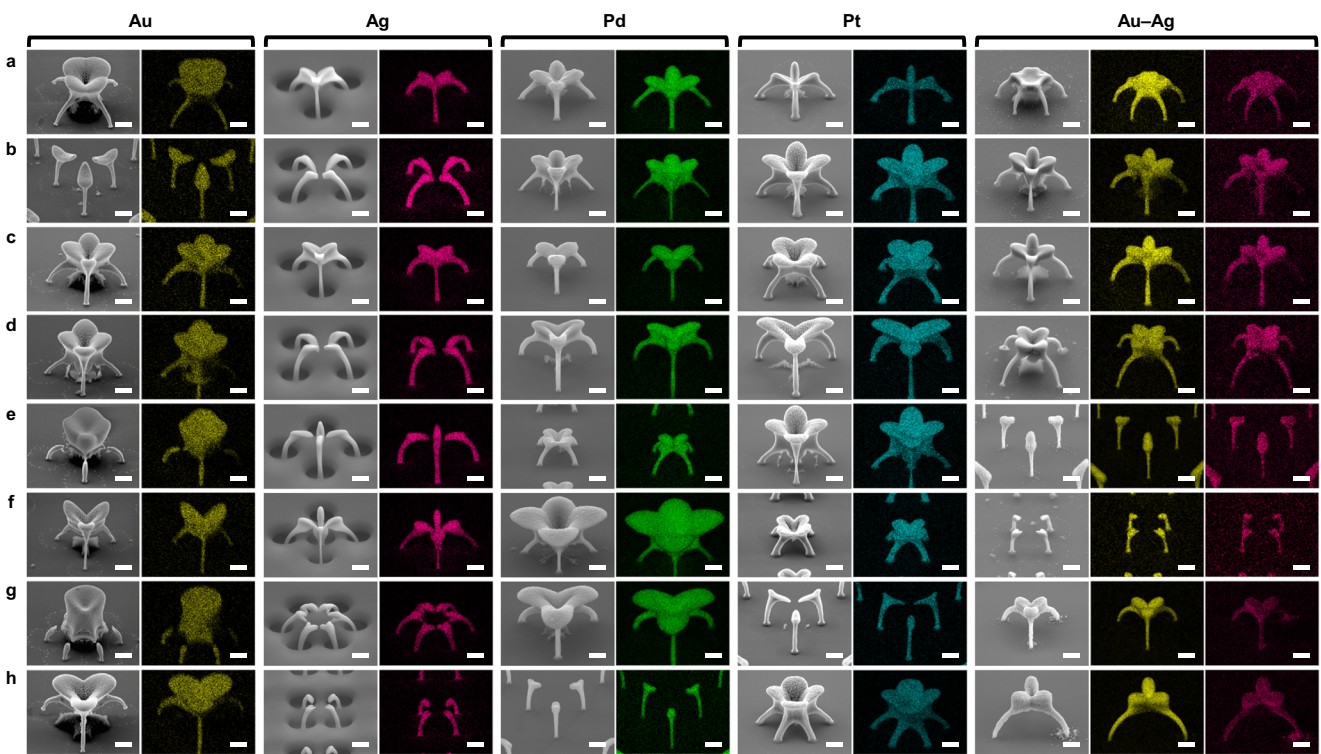

**Fig. 4 | EDS mappings of the 3D nanoarchitectures.** The arrangement of the images is identical to the order used in Fig. 3. Besides the ability for printing arrays, the detailed geometries and compositions for each nanoarchitecture from the arrays are clearly demonstrated. All scale bars are fixed to 1 μm.

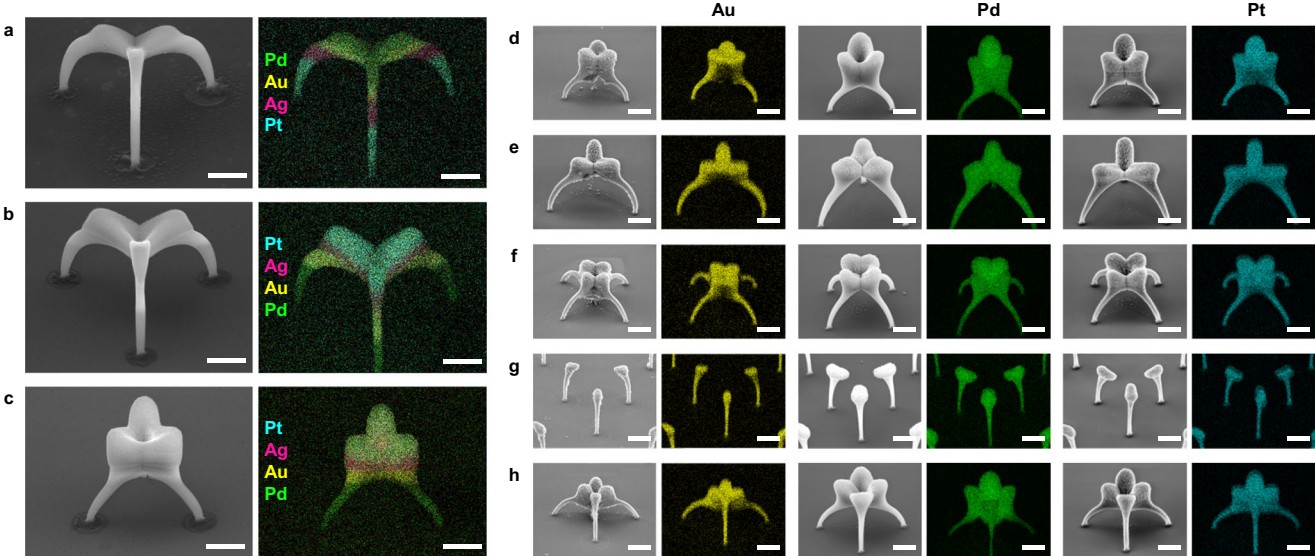

**Fig. 5 | Printing of multiple materials. a–c** Multimaterial (Pt, Ag, Au, and Pd) printing. Despite the reversed order in which the materials occur within the nanoarchitectures (a: Pt–Ag–Au–Pd, b: Pd–Au–Ag–Pt), structural uniformity is maintained. Different nanoarchitectures are also printed by keeping the sequential arrangement of the materials (b, c). Nanoarchitectures printed by combining four aerosol streams for multimaterial printing (more details are provided in Supplementary Fig. 8 and Supplementary Table 3). **d–h** Five individual nanoarchitectures consisting of three different materials (Au, Pd, and Pt), demonstrating the power of our printing strategy: switching materials still results in the successful printing of similar nanoarchitectures. All scale bars represent 1 μm. The tilt angles for SEM imaging are reported in Supplementary Table 1. Pattern designs are presented in Supplementary Table 5.

printing, as the physics concerning the motion of charged NPs is maintained. As a groundbreaking development, we successfully printed 8000 × 8000 uniform 3D nanostructures over an area of 16 mm² within 20 min. Additionally, 200-nm-thick bending nanowires that were printed achieved a high-aspect ratio exceeding 15. To demonstrate nanoscale precision, we printed metal lines that were 14 nm wide. Electric fields could be configured at the atomic scale, thus providing a possibility to assemble smaller building blocks, such as atoms or atomic clusters, rather than particles that were several nanometers in size. This downscaling was supported by selecting sub-2-nm particles for the in situ printing of the 14-nm-wide lines. This strategy for cluster printing is expected to transcend wavelength-dependent techniques. Moreover, the printed area (16 mm²) is nearly three orders of magnitude larger compared to that in previous studies[25–28]. The nanoprinter also enables

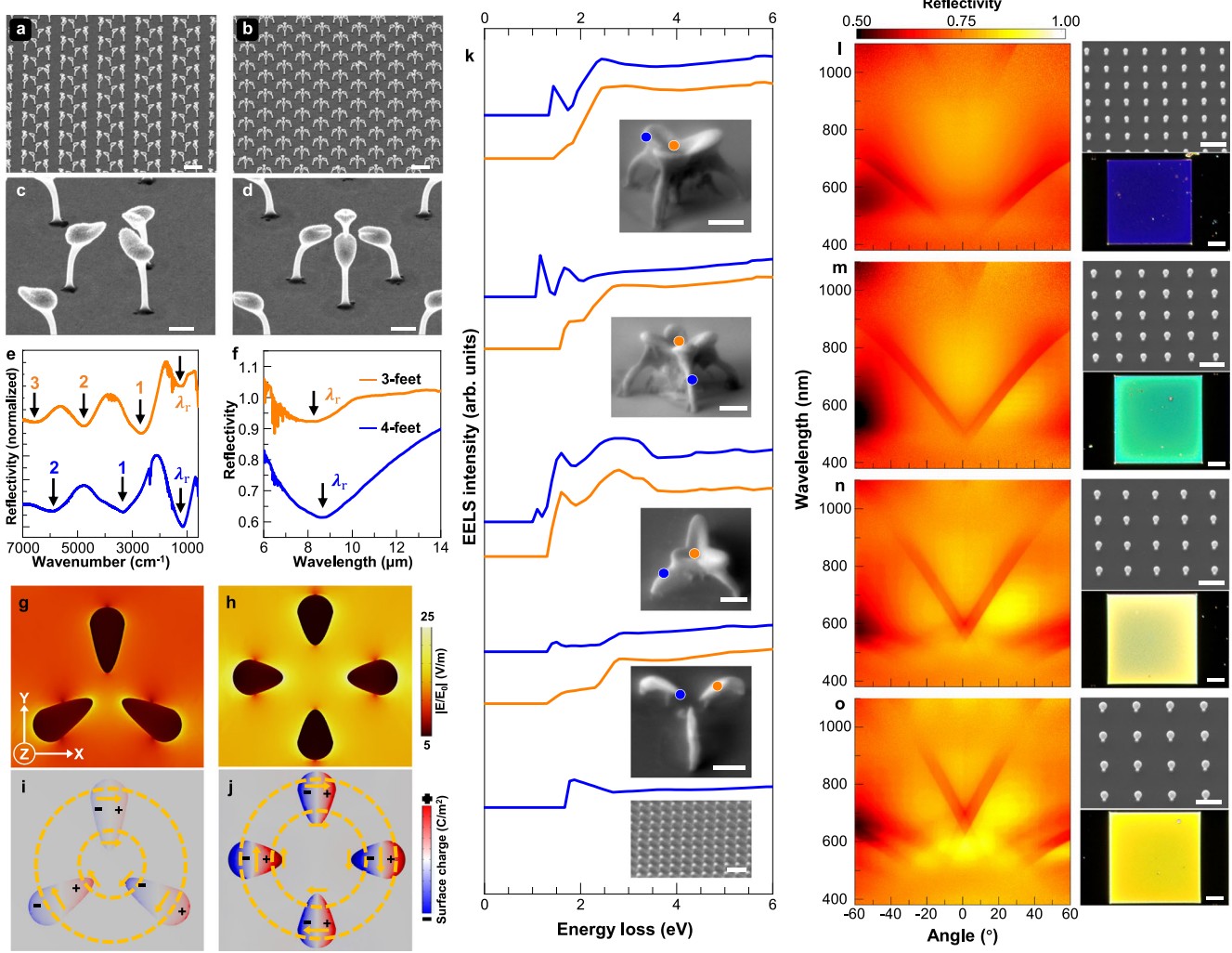

**Fig. 6 | Optical properties of 3D-printed metal nanoarchitectures. a–d** SEM images of the printed "ballet feet"-like structures made of Pd (scale bars in **a**, **b** and **c**, **d** represent 5 and 1 μm, respectively). One array has a primary unit consisting of three "ballet feet," whereas the primary unit of the other array has four "ballet feet." **e**, **f** IR reflectance spectroscopy results for the nanostructure arrays (**a**, **b**), with notable differences in absorption peaks marked by numbers (e.g., 1, 2, 3), at a resonance wavelength of $\lambda_r$. The Cartesian coordinate system in **g** is shared with (**h**–**j**), which show the XY cross-sectional electric-field amplitude (**g**, **h**) and corresponding surface charge density mappings (**i**, **j**) taken at resonance wavelength positions ($\lambda_r$) for the structures consisting of three or four "feet"; the yellow dotted circles and solid arrows schematically illustrate the surface charge flow directions, revealing the dipolar mode characteristics. **k** EELS measurements of the 3D-printed Au nanostructures. The insets depict the SEM images (scale bar: 1 μm) of the measured nanostructures. The electron beam was focused on the locations indicated by orange and blue dots, and the resulting spectra show distinguishable peaks. The bottom spectrum was recorded by focusing the electron beam on an area of ca. 25 μm² of the nanostructural array (bottom inset). **l**–**o** Angle-resolved spectra for the micro-region and optical micrographs (scale bar: 20 μm) for the miniaturized 3D Au nanoarchitectures, as shown in the SEM images (scale bar: 500 nm). The images in different colors and angle-sensitive information are mainly associated with the periodic arrangement of the printed nanostructures. The pitch increases from 400 to 700 nm from **l** to **o**, with a step size of 100 nm. The tilt angles for SEM imaging are provided in Supplementary Table 1. Source data are provided as a Source Data file.

the printing of multiple materials ranging from single metals to multi-materials (in forms such as segments, NP–NP, and atomic mixing). To protect the printed nanostructure arrays for future industrial applications, we successfully covered them with a PR layer (Supplementary Fig. 16). In combination with these developments, we envision that our proposed nanoprinter could be directly integrated into a semiconductor assembly, for example, to enable the one-pass printing of microbumps and/or interconnects with nanometer feature sizes over a wafer scale. The as-printed 3D nanostructures exhibited intriguing optical properties, which were tailored according to their material types, geometries, feature sizes, and periodic arrangement. The nanoprinter is able to accommodate a wide range of metals and has exceptional nanoscale precision. These advantages are expected to promote the application of this nanoprinter in numerous fields, including nanophotonics, nanoelectronics, and microelectromechanical systems.

In future research, industrial/academic collaborations should focus on further miniaturization to sub-3-nm nodes and even to atomic-level 3D arrangement.

## Methods

### Homemade 3D nanoprinter: in situ printing of size-selected nanoparticles

Our homemade 3D nanoprinter consisted of a particle source and a printing chamber. The former was based on spark ablation[30,54], which required a pair of rod electrodes made from metals. The electrodes were arranged to face each other and separated by a gap distance of ca. 1 mm, through which, an inert gas (Ar, purity 99.999%) was flushed with a flow rate of 2.4 lpm (24D7, MKS). The electrodes were connected to an RLC electrical circuit. In this circuit, a constant current source was charging the capacitor that was parallel to the inter-

electrode gap. When the voltage over the capacitor reached a breakdown voltage of the carrier gas (here was Ar), a discharge formed between the electrode gap[55]. As a result, a tiny amount of electrode materials (ca. 0.01 µg per spark)[29] was vaporized to condense into particles of 1–5 nm in diameter (measured by Nano-SMPS, TSI), confirmed by the transmission electron microscopy (JEM-1400plus, JEOL) measurements, as shown in Supplementary Fig. 2. A fraction of them (ca. 1–5%)[37] was charged and these charged NPs were then in situ printed with electric fields downstream as described below. One should note that the Au–Ag alloy nanostructures were printed with their alloy NPs generated by installing a pair of Au and Ag electrodes to the particle source. Similarly, Ni–Cr–Co–Mo–Ag nanostructures were printed from the NP source that used a pair of Ni–Cr–Co–Mo and Ag electrodes. This experiment followed the protocol reported[30], and the alloy nature was also confirmed in literatures[56,57] and the EDS mapping results presented here (Figs. 4 and 5).

### Coupled electric and flow fields

We introduced a second layer of clean gas ($N_2$, purity 99.999%) in parallel to the aerosol flow passing above. The clean gas should be inert ones of high purity for protecting the produced aerosol NPs and printed nanostructures. The gas in the aerosol was mainly selected for generating smaller NPs to enable the printing of compact and uniform nanostructures. Downstream of the particle source (ca. 3 cm apart), we created a planar electric field with a metal plate as the roof while a Si wafer with periodic holes as the floor. Within this space, the printing took place while ruling out the role of material differences. This double-layer flow systems avoided the deposition of NPs onto the PR surface. Therefore, the removal of PR became easier by the means of plasma etching (PE-25, Plasma Etch and PLUTO-T, Plutovac).

### Nanofabrication of patterned substrates

The substrate was based on a Si wafer (n-type doped, crystal surface (110), with a dimension: $10 \times 10 \times 0.5$ mm³). A spinner (WS 650, Mycro) was used to coat a layer of a photoresist (Lor5A, Micro Chem) onto the Si wafer; on top of that, another layer of photoresist (S1805, Dow) was also coated. Each was with a thickness of 500 nm. Otherwise, only a single layer of S1805 with the same thickness was coated. Then we conducted photolithographic processes in Mask aligner (MJB4, SUSS) and in laser direct write lithography (Microwriter ML3, Durham Magneto Optics). Besides, we also sputtered (Q300T D plus, Quorum) a layer of Au with a thickness of 25 nm onto the Si wafer. This metal coating can improve the adhesion of 3D-printed nanostructures, beneficial to the measurements. The detailed pattern designs for the substrates are provided in Supplementary Table 5. The printed nanostructures were electrically connected by a conductive substrate, but adequate for demonstrating their plamonic behaviors[58–63]. Electrical isolation between the nanostructures can probably be achieved by either coating a conducting layer on a nonconductive substrate with subsequent removal of the mentioned layer or directly printing them on the nonconductive substrate via neutralizing the remaining charges over the nanostructures.

For printing 14 nm width lines, the initial step was to control the spark parameters to generate sub-5-nm particles with a breakdown voltage of 1.35 kV, a repetition frequency of 300 Hz, and a capacitance of 1 nF. Subsequently, we selected sub-2-nm particles for printing with a clean-gas flow rate of 1.6 lpm and an aerosol flow rate of 5 lpm with a field strength of 0.4 kV/cm. The photoresist (thickness 60 nm, positive type) was patterned (using EBL, ELS-F125G8) on a silicon wafer with a channel array, having a width of 100 nm, a length of 50 µm and a pitch of 150 nm (Supplementary Fig. 7).

### Characterization of the 3D-printed nanostructures

**Scanning electron microscopy and Focused-ion beam.** The 3D-printed nanostructures were characterized in scanning electron microscopy (JSM-IT500HR/LA and JSM-7800F, JEOL). We used an acceleration voltage of 5–10 kV at a working distance of at least 5 mm. EDS was conducted for elemental analysis. Focused-ion-beam milling (JIB-4700F, JEOL) was conducted for uncovering the interior structures of the 3D-printed architectures.

### Fourier transform infrared spectrometer

An array of the 3D-printed nanostructures was measured by infrared reflectivity in Fourier transform infrared spectrometer (VERTEX 70 v, Bruker) equipped with a microscope (HYPERION, Bruker). The spectra were conducted with a normal incident white light at an area of 100 µm with a wave number ranging from 600 to 8000 cm⁻¹.

### Reflection electron energy-loss spectra

REELS was measured in an Auger electron microscopy (JAMP-9510F, JEOL) with an acceleration voltage of 2 kV, providing information on the elemental and electron states of the printed nanostructures. The data were analyzed by removing the zero-loss peak using the reflected-tail method.

### Angle-resolved spectrum system in micro-region

Angle-resolved spectrum system in micro-region (ARM, ideaoptics) was used to measure the reflection spectra of the 3D-printed nanostructure arrays in a micro-region with a source of white light at a wavelength ranging from 380 to 1100 nm. Reflectance spectra were measured as a function of incident angle of the light source varied from −60 to +60˚. Optical images of the nanostructure arrays were captured by using an optical microscope (BX53M, Olympus). Samples were illuminated with a LED lamp and imaged in reflection mode though a ×20/0.45 NA objective lens.

### COMSOL multiphysics simulations

A 3D electromagnetic Maxwell solver was employed to carry out simulations using COMSOL Multiphysics 6.0 package (wave optics module). A plane-wave light source was used to excite the Au-coated Pd structures printed on a substrate from the top (+Z direction) which was surrounded by perfectly matched layer boundary conditions at Z direction and periodic boundary conditions in XY directions (cf. Supplementary Fig. 14). The periodic conditions were set as follows: 10.51 and 8.44 µm in X- and Y-direction for a three-"ballet feet" structure, respectively; 20 and 10 µm in the X- and Y-direction for a four-"ballet feet" structure, respectively. Electric-field amplitude profiles were extracted from the cross-sectional monitor (positioned on the top of the bead) and the 3D surfaces of the structure. Surface charge density distribution mappings were calculated using the outward normal vector, skin effect, and local electric field[64,65]. The feet shape was constructed by combing a hemisphere, and a parabola[66]. For simulating the 3D maps of electric fields, we used the holey array with such information: the PR thickness was 1 µm; five holes were patterned at the four ends and the cross-point of a plus sign, and the nearest ones had a distance of 2 µm; such five holes formed a primary unit for the periodic arrays with a pitch of 15 µm. We used a field strength of 10,000 Vm⁻¹ and a charge density of $8.15 \times 10^{-6}$ cm⁻².

## Data availability

The authors declare that the data supporting the findings of this study are available within the paper and its supplementary information files. Source data are provided with this paper.

## Code availability

The procedures for performing size selection and its control by the flow and electric fields (using MATLAB) are provided in the Supplementary Information.

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

## Acknowledgements

This work is supported by the Major Research Plan of the National Natural Science Foundation of China (Grant No. 92261102), the ShanghaiTech University faculty startup fund, and funding from the Center for Transformative Science. We would like to thank the support from the Center for High-resolution Electron Microscopy (EM02161943), the Soft Matter Nanofab (SMN180827), the Analytical Instrumentation Center (SPST-AIC10112914), and the Quantum Device Lab from ShanghaiTech University. V.D. acknowledges the support from NRF-2021R1I1A1A01050424. We are grateful to Dr. Yilan Jiang's generous help in measuring REELS. Many thanks also go to idealoptics for providing access to the ARM measurements.

## Author contributions

J.F. conceived and supervised the work. B.L. and J.F. designed the experiments and analyzed the data. S.L. designed the prototype of our homemade 3D nanoprinter and simulated the 3D maps of electric fields. V.D. did the simulations for the three-/four-"ballet feet"-like nanostructures. Y.Y. conducted the experiments for large-area printing. Y.Z. printed the metal lines. J.A. printed Ni–Ti alloys and performed FIB with that. B.L. did the other experiments and designed the layouts with guidance from J.F. B.L. drafted the manuscript and prepared the supplementary materials. J.F. revised the manuscripts. Together with B.L., J.F. wrote the manuscript. All authors participated in the discussions and agreed with the contents of this work.

## Competing interests

J.F., S.L., and B.L. have filed two patent applications related to this work through ShanghaiTech University. PCT/CN2022/098811 (under review) refers to the directed migration of a charged dispersion phase in a gas controlled via the action of an electric field, such that the charged dispersion phase is stacked on a substrate to form a specific micro/nanostructure. 202211492609.6 (under review) refers to the invention of a double-layer flow system for decoupling materials from the printing process. The third patent application (202310981632X, under review) related to this work has been filed by J.F. and Y.Y., referring to the use of a pulsed electric field for large-area printing of uniform 3D nanostructures. All authors declare no other competing interest.
