## [Peer Review File · Nature Communications]

Metal 3D nanoprinting with coupled fieldsREVIEWER COMMENTS

Reviewer #1 (Remarks to the Author):

The manuscript by Liu et al. demonstrates the printing of intricate three-dimensional metal structures from the assembly of charged aerosols on a silicon wafer to which a patterned resist is applied. A key innovation is the use of a double-layer flow whereby one layer carries the aerosol while the other layer, which is adjacent to the substrate, flushes away neutral aerosol particles away from the substrate surface while still allowing those that are charged to follow the field lines and be deposited at precise locations. The net effect is to have an end product that is unobscured by the random attachment of neutral particles to either the substrate or emerging structures. The work is rigorously carried out and has arguably produced the most complex structures ever fabricated using the aerosol method. Although I consider the work to be impressive, there are a number of concerns that I feel the authors must address:

(1) The introduction to the manuscript should more accurately reflect the current state of the field as it relates to 3D imprinting with aerosols. In its current form, it tends to downplay/disregard past achievements to the point that a reader could get the impression that this is an entirely new technique. Revisions should place this work within the context of past achievements (e.g., Nature 2021, 592, 54; Acc. Mater. Res. 2021, 2, 1117; Addit. Manuf. 2021, 48, 102432; Additive Manufacturing 2022, 60, 103206).

(2) The experimental configuration shown in Figure 1a is inverted when compared to that shown in Figure S1. Is it important that the aerosol be in the lower layer? If so, then Figure 1a is misleading. Also, an understanding of why argon is chosen for the aerosol while nitrogen is chosen as the purging gas should be provided.

(3) It is unclear what electrode materials were used when making alloys. When, for example, a AuAg alloy is produced, is the electrode material an alloy or is one electrode Au and the other Ag? If it is the latter, then what is this evidence that a well-mixed alloy is formed in the printed structures as opposed to a composite of Au- and Ag-rich segments.

(4) Claims like "further downscaling becomes easy" and "unprecedented" as they relate to future work should be eliminated from the abstract and conclusion. Stating these points as "possibilities" or "opportunities" seems more appropriate at this stage of development.

(5) The schematic shown in Figure S8a refers to a pulsed printing field which is never described in the manuscript. The schematic of the printing chamber shown seems quite different from the schematic shown in Figure S1. Some clarification is required.

(6) There are essentially no fabrication details provided regarding the 14 nm width lines. Such information should be provided, especially since this is a major claim made in the abstract..

Reviewer #2 (Remarks to the Author):

The manuscript entitled 'Metal 3D nanoprinting with coupled fields' addresses important aspects of the need for structural flexibility in fabricating nanoscale metallic structures and devices. The authors show a very interesting concept using aerosol nanoparticle coupled to a structured electric field to attract charged nanoparticles to land on the specific pattern on the surface. They show several materials and pattern layout and also address scalability in large areas.

The results are very appealing and validate largely the hypotheses addressed in the paper.

The paper however should be improved in following aspects to match the quality standard of a high-impact journal:

- since the work of using aerosol to deposit charged NP is not entirely new and has already been covered to some extent by some of the authors in previous papers, the novelty and specific results aimed at and achieved in this work should be clearly highlighted. When reading the manuscript, links to previous works by the same authors are only mentioned vaguely, leaving the reader to look up old papers and find out the level of print results achieved previously and to

compare with the newest generation. I consider this is an important task to be done by the authors. It won't take away the quality of the work, but would help the reader to see immediately the link to previous, own work and the progression that was made since then.

- the work needs a thorough language check. Even though there are only few typos, some parts are poorly written and hard to understand the meaning behind sentences. As example: " ... "

- it would be good to show some resistivity data of printed metal nanowires/lines.
- it would be good to show some mechanical stability tests of the printed features.

I think the comparison to EUV is out of place as EUV is for high-end, VLSI integration of complex IC, not something addressed here.

- one technical aspect needs to be clarified: ... on page 16 is detailed, that a 25 nm thick Au layer is sputtered onto the silicon to render the surface more adhesive. That means that all printed metal nanostructures are electrically connected through this Au film and the doped Si underneath. If that is correct, is there a way to subsequently isolate the features electrically? Or is it OK to have them connected for the envisioned applications listed in the introduction and motivation?

All in all this is interesting work and nice results, following on the previous work achieved by the group and lead authors. Thus, before this manuscript is ready for publication, it should state clearly the progress compared to the previous papers in more detail (it will make this paper more enjoyable to read). Further it needs to fix some language to be more concise and clear.

Other observations that deserve improvements:

- > it would be helpful to have line numbers for the specific comments.
- > it would be helpful to have all authors listed and not et al.
- > check past tense versus present tense, there is a mix

- can the title be more specific? there is previous work already reporting on the general idea
- abstract: repeats 'nanoarchitectures 2x in one sentence'
- abstract: "intriguing" is not a good scientific term, can you be more specific?
- abstract: "over large areas", it is better to give numbers. Large is relative.
- abstract: "yielding new opportunities evolving to an unprecedented nanomanufacturing platform to advance future nanophotonics and semiconductor devices". This sentence has ZERO meaning, it is just a sales pitch place holder. It should be removed from a journal of this calibre.

Main:

p2

"the development of the nanoworld" doesn't sound very scientific...

" for microelectronics it is necessary to fabricated three-dimensional ...": for what type of microelectronics?

" what is z-axis space" ?

" in a clever demonstration" check word clever is really the best

p3:

" we reflect the mapping of the electric field ..." not clear

p6:

" good mechanical robustness and conductivity": how can you deduce this from the SEM images after the FIB cut alone?

p8:

Text between: "The domain of the ions deposited ... " to " ... such correlations of the two fields" is poorly explained and hard to understand. Try to rephrase it better and clearer.

p9:

There are sentences that start with "presumably", "observably", "interestingly". I think they can all go and doesn't change the sentence. Or use other terms.

Fig 3 has way too many images. In the caption is said "a variety of arrays of geometrically..." but it is not clear what parameters have been used: while the columns are clear (Au, Ag, etc), it is not clear what is the common part in a, b, c, etc. It would be better to rearrange the image and maybe select some representative images and put the rest into the SI.

Fig 4: same for this image... too many small images but it is better to improve viewing quality or say what is in a, b, c, etc.

p 13

the term 'multimaterial' is overstated... or do the authors were able to switch material besides Au/Ag in one print?

what means: "optimal uniformity"

p14

What means: "such a striking method"? check language.

Reviewer #3 (Remarks to the Author):

The manuscript titled "Metal 3D nanoprinting with coupled fields" illustrates a new method of fabricating 3D metal nanoarchitectures by tuning both the electric and flow fields within the self-developed nanoprinter. It is proposed that such a production route has the main advantages of: (1) producing periodic and uniform nanoarchitectures over large areas up to $4 \times 4 \text{ mm}^2$ within a short time frame of 20 mins; (2) having an unprecedented metal line precision of 14 nm; (3) and being able to access a wide range of feedstock materials.

Despite being one of the main advantages repeatedly emphasized by the authors, the evidence of structural uniformity over a large length scale of $4 \times 4 \text{ mm}^2$ is missing. All data within the manuscript are within the range of hundreds of micrometers, with no clear indication of their locations within the overall fabrication platform. It remains questionable to the current reviewer, that as the NP-containing aerosol flow covers a larger distance, will the reduction of "charged NP density" due to printing at the beginning of the aerosol flow subsequently affect the print quality at the end of the aerosol flow.

It is debatable if the "unprecedented 14 nm metal line" precision will make much sense in the context of 3D nanoprinting. Conceivably, with the support of a planar substrate, the metallic bond formation for 2D structures will be much easier comparing to their 3D counterpart, yielding a finer spatial resolution. A more direct / meaningful comparison will be the resolution of 3D structures.

The authors mentioned "multimaterial printing" on several occasions including one subtitle, wherein what they actually meant is a "wider material library". "Multimaterial printing" typically refers to individual structures that are made of by more than one type of material. However, in the current study, all structures are only composed of one material type.

Moreover, it is stated in the manuscript line 270 to 272 that "This achievement (a wide material library) represents an improvement over the state-of-the-art techniques used in the field of microscale 3D printing, most of which are only capable of handling a few types of metals²⁵ and face challenges in printing alloys²⁷." Please discuss and elaborate on the potential reasons or mechanisms behind such an improvement. From the metallurgy perspective, the materials processed in the current study all contain metallic bonding, they should be fairly easy to be consolidated. From the processing perspective, the major changes made by the current work is to control the NP size and local/cushion field, both certainly will influence the structural geometric

resolution, but unlikely to affect atomic bonding. In other words, what is the underlying technical advancement that enables the current nanoprinter to process more metal and alloy types compared other techniques?

The particle size selection criteria (equation 1 in the manuscript) formulated by the authors could potentially serve as an important guideline for any future replication / employment of the current technique. More information should be provided regarding on several important details. (1) How will the NP density within the aerosol flow affect its size selection. (2) How are the parameters within the Cunningham correction factor (e.g., α and β) determined, are they material dependent or process dependent? (3) The validation of the criteria is made with Au NPs (Fig. 1f,g). Did the authors also validate the criteria with other types of NPs to confirm its general applicability?

Is the determination of the characteristic size R (Fig. 2d) made based on 1 material or several materials? This will affect the general applicability of the analytical model.

The labels in Fig. S9 are too small to make any clear interpretation.

For the demonstration of optical properties made by metal nanoarchitectures in Fig. 5, how does the current process compare to the plasmonic resonance metasurface? Have the authors considered any surface protection methods, to prevent potential damages during the actual industrial applications?

We thank all the reviewers for their constructive suggestions and comments, which certainly helped us to improve our manuscript. In the response letter, the font in red represents the reply to the reviewers' comments and *the italic ones are the corresponding changes made to the manuscript and supplementary information*. We have fully addressed all the concerns from the reviewers and the detailed point-to-point responses are provided as below.

Reviewer #1 (Remarks to the Author):

The manuscript by Liu et al. demonstrates the printing of intricate three-dimensional metal structures from the assembly of charged aerosols on a silicon wafer to which a patterned resist is applied. A key innovation is the use of a double-layer flow whereby one layer carries the aerosol while the other layer, which is adjacent to the substrate, flushes away neutral aerosol particles away from the substrate surface while still allowing those that are charged to follow the field lines and be deposited at precise locations. The net effect is to have an end product that is unobscured by the random attachment of neutral particles to either the substrate or emerging structures. The work is rigorously carried out and has arguably produced the most complex structures ever fabricated using the aerosol method. Although I consider the work to be impressive, there are a number of concerns that I feel the authors must address:

(1) The introduction to the manuscript should more accurately reflect the current state of the field as it relates to 3D imprinting with aerosols. In its current form, it tends to downplay/disregard past achievements to the point that a reader could get the impression that this is an entirely new technique. Revisions should place this work within the context of past achievements (e.g., Nature 2021, 592, 54; Acc. Mater. Res. 2021, 2, 1117; Addit. Manuf. 2021, 48, 102432; Additive Manufacturing 2022, 60, 103206).

The reviewer's comments are well taken. Light-matter interaction and electronic devices at nanoscale drives the development of metal 3D nanoprinting. Within this background, we introduced the need to print multiple materials at nanoscale and summarized the most relevant techniques that can enable this possibility. The literatures pointed out by the reviewer are important achievements to show the development of aerosol-based 3D nanoprinting. As suggested by the reviewer, we have added up these literates and the corresponding descriptions read as:

“3D nanoprinting techniques based on aerosols have been considerably developed in different aspects^{1,2,3,4}; nevertheless, the ability to print multimaterials still remains absent, mainly due to its strong dependence on material physics^{5,6}. Four different metals (Au, Ag, Cu and Pd) were printed within one single pillar, but the dimensions of these metals varied from one to another¹, which is likely related to the difficulties in decoupling materials for printing. Besides, these works also printed nanostructures within an area of sub-1-mm² and the smallest feature size was reported as ca. 70–100 nm^{1,2,3,4}, because structural uniformity and miniaturization were often compromised for large-area printing^{1,3,6}.”

(2) The experimental configuration shown in Figure 1a is inverted when compared to that shown in Figure S1. Is it important that the aerosol be in the lower layer? If so, then Figure 1a is misleading. Also, an understanding of why argon is chosen for the aerosol while nitrogen is chosen as the purging gas should be provided.

We thank the reviewer for raising the concern on this configurational arrangement, as this key development not only exerts size selection but also decouples materials in the printing. The aerosol flow should be separated by a layer of clean gas from the substrate, as correctly illustrated in Fig. 1a. One role of the clean gas layer (with a thickness of xx mm) is to avoid the diffusion of neutral particles to the printing area, as the sub-5-nm particles can diffuse violently (associated Brownian displacement lies in the range of 300 μm in 1 s). Another important role for this clean gas layer is to remove the nanoparticle cloud³, where charged and neutral particles are concentrated. This cloud layer ruins the uniformity of the printed nanostructures, as explained previously³. Since we can neglect the gravitational forces, which is only a millionth of the electrostatic force, for these sub-5-nm particles (balance forces are drag and electrical ones), the flip or rotation of the printer does not make any differences. In other words, the clean gas layer only requires to be inserted between the aerosol flow and the substrate surface. For making this claim clearer, we have modified the text in the main manuscript, reading as (on page 5, line 130):

“To guarantee the uniformity of the printed nanostructures, the clean gas layer must be sandwiched between the aerosol flow and the substrate surface for avoiding NP diffusion to the printed area and to remove the NP cloud³.”

For consistency, we also flipped over the schematic in Supplementary Fig. 1:

Supplementary Fig. 1: Schematic of our homemade 3D nanoprinter for coupling the electric and flow fields while exerting the key ability for size selection of NPs and their in-situ printing.

The criterion for selecting the type of clean gas is based on their protection to the aerosol nanoparticles and printed nanostructures. Therefore, the gas should be inert ones of high purity. The reason for choosing argon for spark is to generate smaller nanoparticle building blocks⁷ for ensuring the printing of more compact and uniform nanostructures. The argon has a lower spark energy (proportional to the square of the breakdown voltage) than that of nitrogen, as Paschen law informs that its breakdown voltage is approximately 1/2 of nitrogen under our experimental conditions. For clarifying this, we have modified the text, reading as (on page 18, line 489):

“The clean gas should be inert ones of high purity for protecting the produced aerosol NPs and printed nanostructures. The gas in the aerosol is mainly selected for generating smaller NPs to enable the printing of compact and uniform nanostructures.”

(3) It is unclear what electrode materials were used when making alloys. When, for example, a AuAg alloy is produced, is the electrode material an alloy or is one electrode Au and the other Ag? If it is the latter, then what is this evidence that a well-mixed alloy is formed in the printed structures as opposed to a composite of Au- and Ag-rich segments.

This remark is helpful to make a stronger statement: a pair of electrodes consisting of different materials can create their alloys using spark mashup⁶, without the need of the parent electrodes to be alloyed. This work indeed used Au and Ag electrodes to make their alloy nanoparticles. The mixing mechanism was described in our previous work⁶, which demonstrated the successful synthesis of a variety of alloy nanoparticles (from binary to high-entropy alloys), without showing any element-rich segments. Our work adopted the same particles source for printing, so the printed nanostructures maintained the alloy state of the nanoparticle building blocks. This information was also confirmed by our EDS mapping of the printed Au–Ag alloy nanostructures (cf. Fig. 4, Supplementary Fig. 12). For the convenience of the reviewer, we also moved the HAADF TEM images and their EDX maps for a number of alloy nanoparticles produced in our previous work⁶ to here:

High-angle annular dark field—scanning transmission electron microscopy images and elemental maps of a number of different alloy particles of sub-3-nm in size. (A–F) Binary miscible (PdFe, A) and immiscible (IrCu, B; PtAu, C; AuW, D; CuFe, E; IrAg, F) alloy NPs. (G–I) Ternary (PdPtAu, G), quaternary (CuAgAuPd, H), and senary (AuNiAgCrCoMo, I) alloy NPs.

Editorial Note: Reprinted with permission from Ref. 6. Copyright (2020) Elsevier.

Our previous work⁶ challenged the synthesis of most difficult alloy nanoparticles consisting of immiscible materials in their bulks (e.g., IrCu, PtAu, AuW, CuFe, IrAg),

which were impossibly made by any other method. As Au and Ag are miscible, their alloy nanoparticles were more easily synthesized with a pair of parent electrodes made from Au and Ag, respectively^{8,9}. To claim this, we added the following text in the Method section (on page 18, line 482):

“One should note that the Au–Ag alloy nanostructures were printed with their alloy NPs generated by installing a pair of Au and Ag electrodes to the particle source. This experiment followed the protocol reported⁶, and the alloy nature was also confirmed in literatures^{8,9} and the EDS mapping results presented here (Fig. 4).”

(4) Claims like “further downscaling becomes easy” and “unprecedented” as they relate to future work should be eliminated from the abstract and conclusion. Stating these points as “possibilities” or “opportunities” seems more appropriate at this stage of development.

We have removed these words and modified the text, reading as:

“Electric fields can be configured at the atomic scale, thus providing a possibility to assemble smaller building blocks, such as atoms or atomic clusters, rather than those of particles of several nanometers in size. This downscaling is supported by selecting sub-2-nm particles for in-situ printing of the 14-nm-wide lines. This strategy for cluster printing is expected to exceed the limit from wavelength-dependent techniques. Moreover, the printed area (16 mm²) is scaled up to approximately three orders of magnitude compared to previous works^{1,2,3,4}. The nanoprinter also enables the printing of multiple materials: from single metals to multimaterials (in forms such as segments, NP–NP, and atomic mixing). To protect the printed nanostructure arrays for future industrial applications, we successfully covered them with a PR layer (Supplementary Fig. S16). In combination with these developments, we envision that our proposed nanoprinter can be directly integrated into semiconductor assembly, for example, to enable the one-pass printing of microbumps and/or interconnects with nanometer feature sizes over a wafer scale.”

(5) The schematic shown in Figure S8a refers to a pulsed printing field which is never described in the manuscript. The schematic of the printing chamber shown seems quite different from the schematic shown in Figure S1. Some clarification is required.

We are grateful for this good reminder. Supplementary Fig. 8a was simplified for only illustrating the principle of pulsed electric field in the printing zone, as shown in Supplementary Fig. 1. To avoid misleading, we have extended this schematic to a full setup in Supplementary Fig. 8a, as shown below:

Supplementary Fig. 8: Printing metal 3D nanostructures over large areas. a, Schematics of the nanoprinter. Red area represents the printing area in which a pulsed electrical field was applied. Dashed lines refer to trajectories of NPs with different sizes. The arrows only indicated the direction. **b,** SEM image for showing the areas containing various arrays of the printed nanostructures. The entire area covered $2 \times 1 \text{ mm}^2$. In total, there were 32 squared areas, each of which had the printed nanostructures. The image consisted 8 ones in a row and 4 ones in a column. For printing, the gas flow was designed downward along the image. **c,** SEM image of the printed nanostructures over an area of $4 \times 4 \text{ mm}^2$ on a substrate with a dimension of $5 \times 5 \text{ mm}^2$. Dashed squares marked the three representative locations of the printed nanostructures, whose SEM images are accordingly shown in (d–f). **g–i,** Enlarged views of the SEM images for the nanostructures within the same area as shown in c–e. See Supplementary Discussion 2 for additional information for realizing large area printing. Scale bars: **b, c,** $500 \mu\text{m}$; **d, e, f,** $10 \mu\text{m}$; **g, h, i,** $2 \mu\text{m}$.

The pulsed field was designed to transport smaller charged aerosol nanoparticles downstream and this allows one to print large areas of uniform nanostructures. Without using pulsed field, the charged nanoparticles of different electrical mobilities were printed only within a shorter distance along the aerosol flow ($< 1000 \mu\text{m}$). For a longer distance, the printed nanostructures showed non-uniformity due to the different sized NPs used for printing. With this information, our experiments were properly designed for ensuring the uniform printing. We have added the description in the main manuscript as below:

“The large area printing was achieved by setting a pulsed electric field in the printing zone, because charged aerosol NPs can be distributed over the entire area designed for printing nanostructures (Supplementary Fig. 8).”

(6) There are essentially no fabrication details provided regarding the 14 nm width lines. Such information should be provided, especially since this is a major claim made in the abstract.

We have added the fabrication details for the printing of 14 nm width lines in the Method section, reading as:

“For printing 14 nm width lines, the initial step was to control the spark parameters to generate sub-5-nm particles with a breakdown voltage of 1.35 kV, a repetition frequency of 300 Hz, and a capacitance of 1 nF. Subsequently, we selected sub-2-nm particles for printing with a clean gas flow rate of 1.6 lpm and an aerosol flow rate of 5 lpm with a field strength of 0.4 kV/cm. The photoresist (thickness 60 nm, positive type) was patterned (using EBL, ELS-F125G8) on a silicon wafer with a channel array, having a width of 100 nm, a length of 50 μ m and a pitch of 150 nm (Supplementary Fig. 6).”

We have also added SEM images in Supplementary Fig. 6 and marked the relevant geometrical parameters of the channel pattern used for printing the 14 nm width of the Au lines. In addition, the tilt views showed the line structure as nanowalls with an aspect ratio of 3–5, truly categorized into 3D nanostructures¹⁰.

Supplementary Fig. 6: Nanoscale precision. *a*, SEM image of array nanostructures from Fig. 1j. *b*, An area of multiple line structures corresponding to the SEM image shown in Fig. 1k. The substrate for printing was patterned by an array of channels with a width of 150 nm, and the cracks formed due to sputtering Au layer on the PR surface. *c–e*, SEM images for showing sub-20-nm line width. *f–g*, Tilt views of the lines are actually true 3D structures, called nanowalls with an aspect ratio of approximately 3–5 (height: 60 nm, as marked in the right panel in *g* with a tilt angle of 53° for imaging, width: 24 nm, length: 10 000 nm). Scale bars: *a*, 1 μm; *b*, *c*, *f*, *g*, 100 nm; *d*, 40 nm; *e*, 30 nm.

Reviewer #2 (Remarks to the Author):

The manuscript entitled 'Metal 3D nanoprinting with coupled fields' addresses important aspects of the need for structural flexibility in fabricating nanoscale metallic structures and devices. The authors show a very interesting concept using aerosol NP coupled to a structured electric field to attract charged NPs to land on the specific pattern on the surface. They show several materials and pattern layout and also address scalability in large areas.

The results are very appealing and validate largely the hypotheses addressed in the paper.

The paper however should be improved in following aspects to match the quality standard of a high-impact journal:

- since the work of using aerosol to deposit charged NP is not entirely new and has already be covered to some extent by some of the authors in previous papers, the novelty and specific results aimed at and achieved in this work should be clearly highlighted. When reading the manuscript, links to previous works by the same authors are only mentioned vaguely, leaving the reader to look up old papers and find out the level of print results achieved previously and to compare with the newest generation. I consider this is an important task to be done by the authors. It won't take away the quality of the work, but would help the reader to see immediately the link to previous, own work and the progression that was made since then.

Thank the reviewer for this important reminder. We agree with the reviewer that to summarize the relevant literatures helps to demonstrate the developments achieved here. Our work uniquely leveraged the flow and electric fields to achieve a higher resolution (sub-10-nm, as evidenced by printing 14 nm metal lines), larger areas ($4 \times 4 \text{ mm}^2$) and a wider range of materials (7 types of metals and alloys: Au, Ag, Pd, Pt, Ni-Ti, Au-Ag, Ni-Cr-Co-Mo-Ag, their mixing at different scales and in segmental forms). The previous works successfully printed the smallest pillar diameter of ca. 70 nm within an area of ca. 0.01 mm^2 and showed the controllability to Cu, Pd, and Au nanostructures. For the convenience of the reviewer, we also made a table below to compare the key developments of our work.

Response Table 1. Comparison of our developments with previous works^{1,2,3,4} using aerosol nanotechnology.

	the present work	Previous works^{1,2,3,4}
Printing principles	Coupled fields for size selection	No
Avoiding NP diffusion onto printing areas	Yes	No
Removing the nanoparticle cloud	Yes	No
Decouple materials from the printing	Yes	No
Materials	Metals: Au, Ag, Pd, Pt Binary alloys: Ni–Ti, Au–Ag High-entropy alloy: Ni–Cr–Co–Mo–Ag Multimaterials with in-situ mixing of aerosols: Au, Ag, Pd, Pt Multimaterials with multiple particle sources: Au, Ag, Pd, Pt	Metals: Cu, Pd, Au
Minimum feature size	14 nm	~ 70 nm
Printed area	16 mm ²	0.01 – 0.5 mm ²
Number of nanostructures	64 000 000	~ 3000

For conveying this information, we have modified the main manuscript, reading as:

“3D nanoprinting techniques based on aerosols have been considerably developed in different aspects^{1,2,3,4}; nevertheless, the ability to print multimaterials still remains absent, mainly due to its strong dependence on material physics^{5,6}. Four different metals (Au, Ag, Cu and Pd) were printed within one single pillar, but the dimensions of these metals varied from one to another¹, which is likely related to the difficulties in decoupling materials for printing. Besides, these works also printed nanostructures within an area of sub-1-mm² and the smallest feature size was reported as ca. 70–100 nm^{1,2,3,4}, because structural uniformity and miniaturization were often compromised for large-area printing^{1,3,6}.”

- the work needs a thorough language check. Even though there are only few typos, some parts are poorly written and hard to understand the meaning behind sentences. As example: " ... "

We have thoroughly checked the manuscript and corrected the typos.

- it would be good to show some resistivity data of printed metal nanowires/lines.
- it would be good to show some mechanical stability tests of the printed features.

The reviewer’s suggestions are important to demonstrate the electrical and mechanical properties of the printed nanostructures, which were reported previously¹. In that work, the resistivity of the printed nanostructure was a factor of approximately 2–4 to their

bulk counterparts, and its Young's modulus was approximately 70–90 GPa, comparable to those from their bulks. For delivering these important properties, we have added the description in the main manuscript as below:

“Since the printing took place in an inert gas environment, the NP building blocks remained highly pure⁶. Deposition of such pure NPs led to local coalescence to form compact nanostructures approaching bulk properties, which were confirmed previously to have comparable electrical and mechanical properties to those of their bulk counterparts¹.”

I think the comparison to EUV is out of place as EUV is for high-end, VLSI integration of complex IC, not something addressed here.

We agree with the reviewer that EUV has been a dominating solution for VLSI integration. However, nanolithography has faced the challenges for wavelength-limited critical dimensions. The emergence of EUV pushed Moore's law forward but inadequate to fulfill the requirements for further downscaling, such as to challenge the nanofabrication of sub-2-nm node. Our spotlight here is to eliminate the mentioned downsize limit by miniaturizing the electric fields, thereby bearing with a fundamentally different tool for evolving into atomic scale manufacturing based on a bottom-up strategy. The metal lines with a width of 14 nm fabricated here only required a single step deposition, without the need of materials transfer with multisteps, as required in nanolithography. Furthermore, we also preliminarily showed the possibility to fabricate 3D metal nanostructures over a large area of 16 mm² with a pulsed electric field, likely available for scaling up to wafer scale nanofabrication. To clarify these important developments that can challenge the currently dominating EUV for nanofabrication, we have changed the descriptions as:

“Electric fields can be configured at the atomic scale, thus providing a possibility to assemble smaller building blocks, such as atoms or atomic clusters, rather than those of particles of several nanometers in size. This downscaling is supported by selecting sub-2-nm particles for in-situ printing of the 14-nm-wide lines. This strategy for cluster printing is expected to exceed the limit from wavelength-dependent techniques. Moreover, the printed area (16 mm²) is scaled up to approximately three orders of magnitude compared to previous works^{1,2,3,4}. The nanoprinter also enables the printing of multiple materials: from single metals to multimaterials (in forms such as segments, NP–NP, and atomic mixing). To protect the printed nanostructure arrays for future industrial applications, we successfully covered them with a PR layer (Supplementary Fig. S16). In combination with these developments, we envision that our proposed nanoprinter can be directly integrated into semiconductor assembly, for example, to enable the one-pass printing of microbumps and/or interconnects with nanometer feature sizes over a wafer scale.”

- one technical aspect needs to be clarified: ... on page 16 is detailed, that a 25 nm thick Au layer is sputtered onto the silicon to render the surface more adhesive. That means that all printed metal nanostructures are electrically connected through this Au film and the doped Si underneath. If that is correct, is there a way to subsequently isolate the

features electrically? Or is it OK to have them connected for the envisioned applications listed in the introduction and motivation?

These remarks help to open new topics with the implementation of our 3D printing technique. To construct an electric field, we usually need a conductive substrate. However, an insulating substrate can also work because we have to intensify the field strength that allows field penetration into the aerosol phase. Although this way can construct the fields, the difficulties lie in that the charges over the printed nanostructures cannot be transported. Consequently, the nanostructures generate a repelling effect to the coming charged particles of the same polarity. To neutralize these charges, we propose to switch the polarity of the coming particles synchronizing to the reversing polarity of the fields.

With respect to the electrically connected nanostructures, optoelectronic based devices and absorbing materials favor this feature^{11,12}. For applications related to light-matter interactions, the electrical connection among nanostructures generally weakens the signal. However, the signals were still adequate for these all-metal metasurfaces to demonstrate desired applications^{13,14,15,16}, proving that the printed nanostructures are useful for the envisioned applications. For electrically isolating the printed nanostructures, we propose two possibilities for future investigations:

- a) A conductive layer will be coated on an insulating substrate for printing. Subsequently, this conductive layer will be etched away while keep the printed nanostructures survived with a conducting “foot”.
- b) The nanostructures will be directly printed on an insulating substrate with charge neutralization by polarity reversal.

To bring out these possibilities, we have added the following text in the main manuscript (on page 19, line 510), reading as:

“The printed nanostructures were electrically connected by a conductive substrate, but adequate for demonstrating their plasmonic behaviors^{11,12,13,14,15,16}. Electrical isolation between the nanostructures can probably be achieved either coating a conducting layer on a nonconductive substrate with subsequent removal of the mentioned layer or directly printing them on the nonconductive substrate via neutralizing the remaining charges over the nanostructures.”

All in all this is interesting work and nice results, following on the previous work achieved by the group and lead authors. Thus, before this manuscript is ready for publication, it should state clearly the progress compared to the previous papers in more detail (it will make this paper more enjoyable to read). Further it needs to fix some language to be more concise and clear.

We thank the reviewer again for the compliments. The whole text has been checked thoroughly for adhering to the rule of conciseness and readability. We have also modified the text to state clearly the progress made here as compared to the previous ones, reading as (also replied to the similar comments raised by reviewer 1 above):

“3D nanoprinting techniques based on aerosols have been considerably developed in

different aspects^{1,2,3,4}; nevertheless, the ability to print multimaterials still remains absent, mainly due to its strong dependence on material physics^{5,6}. Four different metals (Au, Ag, Cu and Pd) were printed within one single pillar, but the dimensions of these metals varied from one to another¹, which is likely related to the difficulties in decoupling materials for printing. Besides, these works also printed nanostructures within an area of sub-1-mm² and the smallest feature size was reported as ca. 70–100 nm^{1,2,3,4}, because structural uniformity and miniaturization were often compromised for large-area printing^{1,3,6}.”

Other observations that deserve improvements:

 it would be helpful to have line numbers for the specific comments.

We have added the line numbers to the word version of the manuscript.

 it would be helpful to have all authors listed and not et al.

We apologize for this inconvenience. The journal requires such a reference format that hides the rest of the names when the authors exceed six ones. For the reviewer's convenience, we have also provided a file with the full names for the references, and the specific file was named as “full names of the references”.

 check past tense versus present tense, there is a mix

We have corrected the tense throughout the manuscript.

- can the title be more specific? there is previous work already reporting on the general idea

The previous work did not report the coupling effect of the fields to issue size selection and to decouple materials from the printing, so we request the kind permission to maintain the current title.

- abstract: repeats 'nanoarchitectures 2x in one sentence'

We have modified the sentence as (on page 1 line17):

“By coupling the electric and flow fields, we successfully print metal 3D nanoarchitectures with various periodic arrays and proven uniformity over areas up to 4×4 mm², within 20 min.”

- abstract: "intriguing" is not a good scientific term, can you be more specific?

We have already modified the description to (on page 1 line 22):

“The as-printed 3D nanoarchitectures exhibited optical properties that are tailorable according to the material types, geometries, feature sizes, and periodic arrangement. Here we show that the homemade 3D nanoprinter not only combines metal 3D printing and nanoscale precision, but also decouples materials from the printing process, thereby yielding new opportunities to advance future nanophotonics and semiconductor devices.”

- abstract: "over large areas", it is better to give numbers. Large is relative.

As replied above to another comment from the reviewer, we have already modified it to a more specific description (on page 1 line 17):

“By coupling the electric and flow fields, we successfully print metal 3D

nanoarchitectures with various periodic arrays and proven uniformity over areas up to 4×4 mm², within 20 min.”

- abstract: "yielding new opportunities evolving to an unprecedented nanomanufacturing platform to advance future nanophotonics and semiconductor devices". This sentence has ZERO meaning, it is just a sales pitch place holder. It should be removed from a journal of this calibre.

Upon request, we have deleted this sentence

Main:

p2

"the development of the nanoworld" doesn't sound very scientific...

We have already changed this sentence to (on page 2 line 34):

" Optical metamaterials^{17,18} and nanoplasmonics^{19,20} have yielded a paradigm shift in conventional optics with the development of nanotechnology²¹."

" for microelectronics it is necessary to fabricated three-dimensional ...": for what type of microelectronics? " what is z-axis space" ?

3D metal interconnects serving as the communication paths among transistors are fabricated through photolithography of multi-step material transfer processes. Printing these interconnects in a 3D freedom provides flexibilities to pack the transistors. Additionally, high aspect ratio of the microbumps is also required in the advanced packaging and this requirement is also fulfilled by the 3D nanoprinting technique for maximizing the z-axis ability (i.e., high aspect ratio), unlike those rigid 2D patterning techniques. Researchers have successfully printed transistors^{22,23}, suggesting that 3D nanoprinting can be developed to a platform technology to fabricate key components in microelectronics.

The z-axis space refers to the vertical axis that is perpendicular to the substrate surface and represents the height of the printed nanostructures achieved in our work. This feature can also be exchangeably used with the term of aspect ratio, describing the ability for controlling the 3D geometries in printing. Our purpose for using this term was to bridge the 3D packaging and the 3D freedom of fabricating nanostructures with our proposed technique.

To clarify these points, we have added the description in the main manuscript (page 2, line 42) as below:

“Moreover, for microelectronics, it is necessary to fabricate three-dimensional (3D) metallic materials occupying z-axis space while maintaining the miniaturized areal dimension, which remains challenging, especially at small scales^{24,25,26}. The capability to control the height of the nanostructures along the z-axis space can be realized by 3D nanoprinting techniques, in contrast to lithographic and 2D material synthesis techniques. Structural flexibilities fulfil the requirements in advanced packaging, and such a freedom likely propels new developments in high density and aspect-ratio microbumps, fine interconnects, and transistors^{22,23,27,28,29,30}.”

" in a clever demonstration" check word clever is really the best
We have already changed the sentence to (on page 2 line 57):

" In a recent demonstration³¹, a laser source was used to shrink a hydrogel framework for printing multimaterials with purity being the primary concern. "

p3:

" we reflect the mapping of the electric field ..." not clear

Electric fields are structured in a 3D form; thus, their spatial distributions can be portrayed as a 3D map. We have changed the text in the main manuscript (on page 3 line 81), reading as:

"Considering that uniform fields can be maintained with the aid of the double-layer flow, we reflect the spatial distribution of the electric field with the geometries of the printed nanostructures. We therefore call this spatial distribution as the field mappings hereafter."

p6:

" good mechanical robustness and conductivity": how can you deduce this from the SEM images after the FIB cut alone?

We appreciate this scientific view, since the selected nanoparticles of sub-5-nm generally proceed with local coalescence upon deposition, mimicking grain growth in metallurgy³². This statement is experimentally supported by imaging the grains of the printed nanostructure after FIB processing (Supplementary Fig. 5i, j, k). The printed nanostructures should therefore have good mechanical strength, and this claim is supported by a number of reports^{33,34,35,36}. The avoidance of pores within the nanostructure leads to approaching bulk resistivity. We have added the description in the main manuscript as below:

"To confirm that the materials were densely packed inside the structure, focused ion beam (FIB) was employed to investigate the interior of the printed Ni–Ti nanostructure (Fig. 1h), and the results showed no porosity. Compared to those with porosity, improved mechanical and electrical properties were reported^{33,34,35,36}."

We also conducted additional experiments for using TEM to image the grains of the printed nanostructure. This information is provided in newly added images to Supplementary Fig. 5, as shown below:

Supplementary Fig. 5: Interior nanostructures after FIB milling. *a–h*, SEM images of Au–Ag nanostructures before and after focused ion beam, showing that the interior structure is densely packed. Red dashed lines and arrow represent the initial place for FIB milling. *i–j*, SEM image (*i*) and TEM images (*j*, *k*) of the printed nanostructures before and after FIB milling, the dashed frames in *k* indicate that the particles have coalesced into larger grains. Scale bar for (*a*, *b*, *d*, *f*, *i*) is 1 μm , for (*c*, *e*, *g*, *h*) is 500 nm, for *j* is 100 nm and for *k* is 50 nm.

p8:

Text between: "The domain of the ions deposited ... " to " ... such correlations of the two fields" is poorly explained and hard to understand. Try to rephrase it better and clearer.

The reviewer's comments were well received. To improve the readability, we have

modified the text accordingly and redrawn the schematic (Fig. 2a) for clearly illustrating the model, reading as:

“The domain of the ions deposited on the PR surface form a local field around the opening patterns. This local field can be imagined as a cushion, and it competes with an externally applied constant field to form a semi-circular shape (Fig. 2a). The neighboring two half-circles squeeze the external-field lines into an electric funnel³ that forms the guiding path for the charged NPs (Fig. 2a). Here we wish to quantify the correlations between the two fields.”

p9:

There are sentences that start with "presumably", "observably", "interestingly". I think they can all go and doesn't change the sentence. Or use other terms.

Following these suggestions, we have removed all these words.

Fig 3 has way too many images. In the caption is said "a variety of arrays of geometrically..." but it is not clear what parameters have been used: while the columns are clear (Au, Ag, etc), it is not clear what is the common part in a, b, c, etc. It would be better to rearrange the image and maybe select some representative images and put the rest into the SI. Fig 4: same for this image... too many small images but it is better to improve viewing quality or say what is in a, b, c, etc.

We agree to this suggestion. Images in Figs. 3 and 4 represent intricate nanostructures made of a variety of different materials and the figures were designed to demonstrate the full control over the materials and architectures with our technique. Fig. 4 arranged the enlarged views of individual nanostructures that are corresponding to the arrays from Fig. 3. Following the reviewer's suggestion, we have detailed the descriptions for each figure. The caption for Fig. 3 reads as:

“Fig. 3: Periodic arrays of 3D-printed metal nanoarchitectures. a–h, SEM images presenting a variety of arrays of geometrically different 3D nanostructures comprising Au, Ag, Pd, Pt and Au–Ag, as indicated above each column, respectively. The 3D nanoprinter allows the printing of different geometries, despite the material differences. Herein, the scale bar is kept constant at 5 μm . The enlarged views of a single nanoarchitecture from the corresponding metallic arrays are presented in Fig. 4 to detail the intricate architectures and materials. The tilt angles for SEM imaging are reported in Supplementary Table 1. Pattern designs for the substrates are presented in detail in Supplementary Table 4.”

The caption of Fig. 4 has also been modified to:

“Fig. 4: EDS mappings of the 3D nanoarchitectures. The image arrangement (a–h in the top panel, bordered by a dashed line) is identical to the order used in Fig. 3. i–k, Multimaterial (Pt, Ag, Au and Pd) printing. Despite the change of material orders within the nanoarchitectures (i: Pt–Ag–Au–Pd, j: Pd–Au–Ag–Pt), structural uniformity is maintained. By keeping sequential arrangement of the materials, different nanoarchitectures are also printed (i, j). Moreover, a different strategy for the multimaterials printing is also achieved by mixing four aerosol streams (more details

are provided in Supplementary Fig. 12 and Supplementary Table 5). **l–p**, Three different materials (Au, Pd, and Pt) printed to a variety of similar nanoarchitectures, demonstrating the power of our printing strategy: switching materials still results in successful printing of similar nanoarchitectures. This important feature is still valid even for printing multimaterials, as identical nanostructures are observed for *k* and *l*. All scale bars are set at 1 μm . The tilt angles for SEM imaging are reported in Supplementary Table 1. Pattern designs are presented in detail in Supplementary Table 4.”

The newly added table can be found in SI, as shown below:

Supplementary Table 4. Pattern designs for the substrates used for printing the structures shown in Figs. 3 and 4. The hole diameter in the pattern was fixed to 800 nm, which can be treated as a scale bar for gaining more information about the geometrical designs of the patterned substrates.

	Au	Ag	Pd	Pt	Au–Ag
a					
b					
c					
d					
e					

p 13

the term 'multimaterial' is overstated... or do the authors were able to switch material besides Au/Ag in one print?

The reviewer is right that we can easily switch materials in one print. To demonstrate this multimaterials ability, we have experimented a number of different nanostructures consisting of multimaterials:

- 1) In one print, we used a particle source for printing a while and then switched to another particle source for sequential printing. By repeating this procedure for four different materials, we then realized the printing of Au—Ag—Pt—Pd in one print. The order arrangement of these materials was also changed but still maintained the structural uniformity.
- 2) The multimaterial printing was also demonstrated in a form of particle-particle mixing, as we just run different particle sources in parallel and then mixed their aerosol phases prior to printing.
- 3) We also achieved the printing of multimaterials in atomic mixing, i.e., printing alloys.

For demonstrating the great flexibility in multimaterials printing of various forms, we have modified the main manuscript, reading as:

“As a proof of concept, the results in Figs. 1m and n indicate that the mapping of the fields is flexibly controlled for printing periodic arrays of nanoarchitectures consisting of multimaterials (printed in the order Pt, Ag, Au and Pd from the top to bottom of the nanostructure), as confirmed by energy-dispersive X-ray spectroscopy (EDS) mapping (Figs. 1n, o). Additional results about multimaterials printing are presented in Supplementary Fig. 12 and Supplementary Table 5.”

We have also provided the key feature of our nanoprinter for multimaterial printing in Fig. 1m-o, reading as:

Fig.1: Coupled electric and flow fields in 3D nanoprining. *a*, Schematic of the homemade 3D nanopriner with a clean-gas layer that is parallel to an aerosol flow. Dashed curves represent the electric-field lines, and the green dots denote charged and neutral NPs. To guarantee the uniformity of the printed nanostructures, the clean gas layer must be sandwiched between the aerosol flow and the substrate surface to avoid NP diffusion to the printed area and to remove the NP cloud³. *b*, Simulation results for the 3D mappings of electric fields in periodic patterns and an enlarged view (*c*). *d*, *e*, SEM images of a 3D-printed flowery 3D nanoarchitecture (scale bar 1 μm) and the corresponding array (scale bar: 5 μm) defined by 3D mapping of the fields. *f*, Particle-size distribution of Au NPs with a grey shadow averaged over repeatable measurements performed five times. The area under the lognormal curve represents the total concentration of NPs, each of which carries a positive elementary charge. The orange and the light-blue areas underneath the lognormal curve represent the two different total concentrations of NPs after size selection according to (*g*). *g*, Particle size (D_p) as a function of the ratio between the electric-field strength and flow rate of the clean gas (lpm: liter per minute). The light-green region indicates the variation of geometric volume of the nanopriner (from the lower to upper edge, the volume reduces by approximately 53%). *h*, SEM images of a structure before and after focused ion beam (FIB) processing (scale bar 500 nm), showing that the interior is densely packed. SEM images for high-aspect ratio (=15) bending nanowires (*i*) (diameter: 200 nm; length: 3000 nm; bending angle: 105°), which can be considered as metasurface in 3D forms. *j*, *k*, *l*, Demonstrated nanoscale precision. A Pt nanostructure with the minimum feature size of 25 nm (*j*) and printing 14-nm-wide Au line (*k*, *l*). The scale bars are 100 nm for (*i*, *k*), 1 μm for (*j*, *l*, *n*, *o*) and 5 μm for (*m*). *m*, *n*, Multimaterial (Pt, Ag, Au and Pd) printing within a single nanostructure, as confirmed with energy-dispersive X-ray spectroscopy (EDS) mapping in (*o*). The tilt angles used for SEM imaging are presented in Supplementary Table 1.

In addition, we have provided new experimental results for demonstrating the ability for multimaterials printing in Fig. 4i–k, Supplementary Fig. 12 and Supplementary Table 5. The corresponding results have been provided as below:

“We also managed to print multimaterials nanoarchitectures (Fig. 4i–k, Supplementary Fig. 12). One can see that changing material arrangement within the single nanoarchitecture does not render structural variations (Fig. 4i, j). Compared with the printing of a variety of single materials, multimaterial printing maintains the desired nanoarchitectures (Fig. 4k, l). In situ mixing of different aerosol phases allows instantaneous switching to a broad range of materials easily, which is an important feature unobtainable in conventional multimaterials printing³⁷. In our work, the aerosols are mixed using four different NP sources (Au–Pt–Ag–Pd), which are then transported to the printing zone to demonstrate a different strategy for realizing multimaterials printing (Supplementary Fig. 12). This strategy is achieved via NP–NP mixing, while the multimaterials printing is also demonstrated in segmental forms within a single nanoarchitecture (Fig. 4i–k) and in atomic mixing (i.e., alloys indicated in the Au–Ag columns of Figs. 3 and 4a–h, Supplementary Fig. 12).”

Fig. 4: EDS mappings of the 3D nanoarchitectures. The image arrangement (a–h in the top panel, bordered by a dashed line) is identical to the order used in Fig. 3. i–k, Multimaterial (Pt, Ag, Au and Pd) printing. Despite the change of material orders within the nanoarchitectures (i: Pt–Ag–Au–Pd, j: Pd–Au–Ag–Pt), structural uniformity is maintained. By keeping sequential arrangement of the materials, different nanoarchitectures are also printed (i, j). Moreover, a different strategy for the multimaterials printing is also achieved by mixing four aerosol streams (more details are provided in Supplementary Fig. 12 and Supplementary Table 5). l–p, Three different materials (Au, Pd, and Pt) printed to a variety of similar nanoarchitectures, demonstrating the power of our printing strategy: switching materials still results in successful printing of similar nanoarchitectures. This important feature is still valid even for printing multimaterials, as identical nanostructures are observed for k and l. All scale bars are set at 1 μm . The tilt angles for SEM imaging are reported in Supplementary Table 1. Pattern designs are presented in detail in Supplementary Table 4.

For highlighting the ability for multimaterial printing, we have moved the results for Au–Ag alloy nanostructures to Supplementary Fig. 12 as below:

Supplementary Fig. 12: Multimaterials printing. Materials differences are marked in colors. **a**, Multimaterial printing: Atomic mixing. A pair of different electrodes was used in spark and generated alloy nanoparticles, which were subsequently printed. **b**, **c**, Mapping of the fields is flexibly controlled for printing periodic arrays of “ballet feet”-like structures made of an Au–Ag alloy, as confirmed by electron diffraction spectroscopy mapping for each element (**d**, **e**). Scale bar: **b**, 5 μm ; **c**, **d**, **e**, 1 μm . **f**, Multimaterial printing: Nanoparticle–nanoparticle mixing. In-situ mixing of different aerosols from multiple particle sources run in parallel to form a mixed aerosol for subsequent printing. **h**, Multimaterial printing: segmented layers consisting of different materials of Pt, Ag, Au and Pd. Switching the particle source in chronological order to print multimaterial nanostructures in segmental forms. **g**, **i**, SEM images and the corresponding EDS mapping for the printed multimaterial nanostructures.

Supplementary Table 5 Experimental parameters for printing multimaterials nanostructures

		Pd	Au	Ag	Pt		
Multi-material printing: NP-NP mixing	Particle source	Breakdown voltage (kV)	1.4	1.6	1.6	1.5	
		Frequency (Hz)	170	490	700	140	
		Capacitance (nF)	1	1	2	1	
	Printing conditions	Clean gas flow rate (Lpm)			1.2		
		Aerosol flow rate (Lpm)			2		
		Field strength (kV cm ⁻¹)			0.625		
		Printing time (min)			240		
	Multimaterials printing: segmental layers consisting of different materials (in the order of Pd- Au- Ag- Pt arranged upward)	Particle source	Breakdown voltage (kV)	1.5	1.7	1.8	1.4
			Frequency (Hz)	390	340	600	430
			Capacitance (nF)	1	1	2	1
Printing conditions		Clean gas flow rate (Lpm)			1.2		
		Aerosol flow rate (Lpm)			2		
		Field strength (kV cm ⁻¹)			0.625		
		Printing time (min)	50	60	240	50	
Multimaterials printing: segmental layers consisting of different materials (in the order of Pt- Ag- Au- Pd arranged upward)		Particle source	Breakdown voltage (kV)	1.6	1.7	1.9	1.4
			Frequency (Hz)	360	340	590	420
			Capacitance (nF)	1	1	2	1
	Printing	Clean gas			1.2		

conditions	flow rate (Lpm)
Aerosol flow rate (Lpm)	2
Field strength (kV cm ⁻¹)	0.625
Printing time (min)	50 90 180 50

what means: "optimal uniformity"

We have changed this to:

“Thus far, we have demonstrated the ability of multiple materials printing with various nanoarchitectures and proven uniformity in large arrays.”

p14

What means: "such a striking method"? check language.

Following the suggestion, we have modified the text to:

“This cluster-assembled concept with fields is expected to exceed the limit from wavelength-dependent techniques.”

Reviewer #3 (Remarks to the Author):

The manuscript titled “Metal 3D nanoprinting with coupled fields” illustrates a new method of fabricating 3D metal nanoarchitectures by tuning both the electric and flow fields within the self-developed nanoprinter. It is proposed that such a production route has the main advantages of: (1) producing periodic and uniform nanoarchitectures over large areas up to $4 \times 4 \text{ mm}^2$ within a short time frame of 20 mins; (2) having an unprecedented metal line precision of 14 nm; (3) and being able to access a wide range of feedstock materials.

Despite being one of the main advantages repeatedly emphasized by the authors, the evidence of structural uniformity over a large length scale of $4 \times 4 \text{ mm}^2$ is missing. All data within the manuscript are within the range of hundreds of micrometers, with no clear indication of their locations within the overall fabrication platform.

We are grateful for this reminder. As an overall area for the printed nanostructures cannot provide the exact geometries, we therefore enlarged the view in a range of hundreds of micrometers. For the printing results of a large length scale of $4 \times 4 \text{ mm}^2$, we provided the SEM images in the SI (Supplementary Fig. 8, Supplementary Discussion 2). To show the (relatively) structural uniformity across the large area, the SEM images (Supplementary Fig. 8 d–i) of the printed 3D nanostructures were taken at different locations along the flow direction, as marked by the dashed squares, whose SEM images were also provided. In contrast to the previous works^{1,2,3,4} for printing a maximum length scale of $0.75 \times 0.75 \text{ mm}^2$, the advancement of our work showed large-area printing over $4 \times 4 \text{ mm}^2$, approximately 30 times of those achieved previously^{1,2,3,4}. We have added these images into Supplementary Fig. 8, in which the full arrays ($2 \times 1 \text{ mm}^2$) of the printed structures have been newly added as Supplementary Fig. 8b.

Supplementary Fig. 8: Printing metal 3D nanostructures over large areas. a, Schematics of the nanoprinter. Red area represents the printing area in which a pulsed electrical field was applied. Dashed lines refer to trajectories of NPs with different sizes. The arrows only indicated the direction. **b,** SEM image for showing the areas containing various arrays of the printed nanostructures. The entire area covered $2 \times 1 \text{ mm}^2$. In total, there were 32 squared areas, each of which had the printed nanostructures. The image consisted 8 ones in a row and 4 ones in a column. For printing, the gas flow was designed downward along the image. **c,** SEM image of the printed nanostructures over an area of $4 \times 4 \text{ mm}^2$ on a substrate with a dimension of $5 \times 5 \text{ mm}^2$. Dashed squares marked the three representative locations of the printed nanostructures, whose SEM images are accordingly shown in **d–f**. **g–i,** Enlarged views of the SEM images for the nanostructures within the same area as shown in **c–e**. See Supplementary Discussion 2 for additional information for realizing large area printing. Scale bars: **b, c,** $500 \mu\text{m}$; **d, e, f,** $10 \mu\text{m}$; **g, h, i,** $2 \mu\text{m}$.

It remains questionable to the current reviewer, that as the NP-containing aerosol flow covers a larger distance, will the reduction of “charged NP density” due to printing at the beginning of the aerosol flow subsequently affect the print quality at the end of the aerosol flow.

We thank the reviewer to mention this mobility-determined size selection in printing. The charged nanoparticles of different electrical mobilities were selected and printed within a distance of $< 1000 \mu\text{m}$ along the aerosol flow with a DC field. For a longer distance, the printed nanostructures started to have non-uniformity. The reason for this is that the smaller nanoparticles were consumed first, while the larger ones were printed downstream (i.e., at the end of the aerosol flow). This non-uniformity problem was addressed by setting a pulsed field during printing. The essence of this setting is to redistribute the charged nanoparticles of different sizes that can entirely cover the printed

area. The results for large-area printing were experimentally verified and presented in Supplementary Fig. 8. As we are filing this large-area printing to EU/USA/China patents, we have briefly described this operation, reading as:

“The large area printing was achieved by setting a pulsed electric field in the printing zone, because charged aerosol NPs can be distributed over the entire area designed for printing nanostructures (Supplementary Fig. 8).”

It is debatable if the “unprecedented 14 nm metal line” precision will make much sense in the context of 3D nanoprining. Conceivably, with the support of a planar substrate, the metallic bond formation for 2D structures will be much easier comparing to their 3D counterpart, yielding a finer spatial resolution. A more direct / meaningful comparison will be the resolution of 3D structures.

The review’s comments are well taken. Line width determines the minimal lateral dimensions that can be printed accurately and reliably, and therefore it has a direct impact on the resolution and precision of the subsequently printed 3D nanostructures. A great number of literatures^{27, 31,38,39, 40,41,42,43,44,45,46} also referred to the line width to demonstrate their printing resolution. This minimal lateral dimensions become a traditional standard for comparing with each technique. One possible reason for this lies in that all the techniques are capable of printing line structures, but uneasy for unifying another identical geometry for comparison. In our system, the high-precision printing is imparted by controlling the nanoscale electric fields around the photoresist patterns and the nanoparticle building blocks. The printing of metal lines with 14 nm width demonstrates a remarkable ability for printing fine (sub-20-nm) features of hard materials, which to our best knowledge, has remained unachievable until now and represent approximately three orders of magnitude advancement as compared to other 3D nanoprining techniques^{37,47,48}. We also would like to mention that the 2D structures act as the “root” of subsequently developed 3D ones.

It is also worthy to point out that the lines printed in our work should be more properly called nanowalls, which had an aspect ratio of approximately 3–5, thus being truly categorized into the 3D nanostructures¹⁰. As agreed in literatures¹⁰, the tradition in 3D printing informs that aspect ratio > 2 is considered as 3D nanostructures. Supplementary Fig. 6 also showed the metal nanowalls with a width and height of approximately 20 and 60 nm, respectively. The reason for us to still use the line width is for easy comparisons with the most advanced precisions demonstrated by other nanofabrication techniques^{10,49}. With that comparison, we are faithful to claim that the printing of metal lines with a width of 14 nm, despite the aspect ratio of ca. 3–5, represents a record-breaking milestone. Another technical reason for using the line width lies in that the bird-view in SEM imaging of the nanowalls appears as lines. The tilt view has also been provided to prove their nanowall structures in Supplementary Fig. 6. To bring this information out, we have added the description in the main manuscript as below:

“The printing of 14-nm wide lines demonstrates the powerful ability of our technique to challenge the minimal lateral dimensions of metals, and laid the foundation for

subsequent printing of complex 3D nanostructures while maintaining the resolution demonstrated. Technically speaking, the metal lines should be more accurately called nanowalls (Supplementary Fig. 6), as they had an aspect ratio of ca. 3 (Supplementary Fig. 6), with a height, width and length of 60, 24, and 10 000 nm, respectively. For the sake of comparison and consistency to the literatures, we use the standard terminology of line width throughout this manuscript.”

As shown in Supplementary Fig. S6, the tilt views showed the line structure as nanowalls with an aspect ratio of 3–5, truly categorized into 3D nanostructures¹⁰. For easy access, we also copied Fig. S6 as follows:

Supplementary Fig. 6: Nanoscale precision. *a*, SEM image of array nanostructures from Fig. 1j. *b*, An area of multiple line structures corresponding to the SEM image shown in Fig. 1k. The substrate for printing was patterned by an array of channels with a width of 150 nm, and the cracks formed due to sputtering Au layer on the PR surface. *c–e*, SEM images for showing sub-20-nm line width. *f–g*, Tilt views of the lines are actually true 3D structures, called nanowalls with an aspect ratio of approximately 3–5 (height: 60 nm, as marked in the right panel in *g* with a tilt angle of 53° for imaging, width: 24 nm, length: 10 000 nm). Scale bars: *a*, 1 μm; *b*, *c*, *f*, *g*, 100 nm; *d*, 40 nm; *e*, 30 nm.

The authors mentioned “multimaterial printing” on several occasions including one subtitle, wherein what they actually meant is a “wider material library”. “Multimaterial printing” typically refers to individual structures that are made of by more than one type of material. However, in the current study, all structures are only composed of one material type.

We thank the reviewer for correcting our descriptions about multimaterials. Indeed, we can switch materials in one print for fabricating multimaterial within one structure. To demonstrate this capability, we did additional experiments as described below:

- 1) We managed to switch materials in one print. We used a particle source for the printing and then switched to another particle source for a sequential printing. We then repeated this switching to particle sources by four different materials (Au–Ag–Pt–Pd). The printing of such multimaterials within a single nanostructure was realized. Despite the change of the orders for material arrangement, the structural uniformity is maintained.
- 2) The multimaterial printing has also been demonstrated in a form of particle–particle mixing, as we just run different particle sources in parallel and then in-situ mix them in aerosol phases prior to printing.
- 3) We also have achieved the printing of multimaterials in atomic mixing, i.e., printing alloys.

For demonstrating the great flexibility in multimaterials printing of various forms, we have modified the main manuscript, and added new images to Fig. 1m–o, Fig. 4i–k, Supplementary Fig. 12 and Supplementary Table 5. Since these results were provided above to reply to a similar comment raised by the reviewer 2, we invite the reviewer to kindly scroll up to pages 18–23 of this response letter for a check.

Moreover, it is stated in the manuscript line 270 to 272 that “This achievement (a wide material library) represents an improvement over the state-of-the-art techniques used in the field of microscale 3D printing, most of which are only capable of handling a few types of metals²⁵ and face challenges in printing alloys²⁷.” Please discuss and elaborate on the potential reasons or mechanisms behind such an improvement. From the metallurgy perspective, the materials processed in the current study all contain metallic bonding, they should be fairly easy to be consolidated. From the processing perspective, the major changes made by the current work is to control the NP size and local/cushion field, both certainly will influence the structural geometric resolution, but unlikely to affect atomic bonding. In other words, what is the underlying technical advancement that enables the current nanoprinter to process more metal and alloy types compared other techniques?

The reviewer’s comments help to highlight the remarkable ability of our technique for printing multiple metals/alloys and multimaterials. However, it should be pointed out that the metal printing, especially at the nanoscale, differs greatly from traditional metallurgy¹⁰. The composition and nanostructure have a decisive influence on the material properties. Emerging applications require the development of new materials and their compositional and nanostructural optimization. Materials discovery and

optimization have hampered these developments. The Edisonian trial-and-error process is time consuming and resource inefficient, particularly when contrasted with vast materials design spaces. Whereas traditional combinatorial deposition methods can generate material libraries, these suffer from limited material options and inability to leverage major breakthroughs in nanomaterial synthesis.

According to the material synthesis principles, the microscale printing techniques are categorized into two main groups¹⁰: metal transfer and in situ synthesis techniques.

The techniques based on metal transfer require the previous synthesis of metallic materials before the actual printing process, while the subsequent deposition simply transfers the pre-synthesized material to the location of interest. This metal transfer is further divided into transfer of nanoparticles and transfer of melt droplets. As a consensus in literatures¹⁰, the synthesis techniques are strongly material dependent.

The in-situ methods rely on the synthesis of the metal at the location of interest during the printing process and its subgroups are chemical reduction and dissociation of metal precursors.

The synthesis and transfer strategies largely vary among the printing techniques for processing different materials. No single synthetic strategy can process multiple materials without other limiting factors. For example, those based on the transport of metal nanoparticles in solution require stabilizers to avoid aggregation (detrimental for making compact structures and for impacting fluidity and rheology). The stabilizers must not influence the purity of the nanoparticles, because otherwise the printed structure is contaminated. These wet-methods often suffer from the impurities and require tedious post-processing procedures, thereby impacting the universality of the printing technique. The key obstacles for processing multiple materials for metal nanoprinting are summarized as below:

- a) The type of materials is often restricted by printing principles, such as great differences for nanoparticle transfer or post-processing.
- b) Precursors are mostly lack, because the conversion to desired metals requires rigorous chemistry reduction and photosensitive reactions. For instance, reduction potentials generally vary from one metal to another, raising more difficulties to make alloys of desired compositions. Similarly, photosensitive materials may not be always present for the target metals.
- c) Impurity has been a big problem in wet-methods, which can be detrimental to form pure metals at nanoscale.
- d) Alloys and high-entropy alloys are not compatible with some nanoprinting techniques, thus forbidding the possibility to be printed.

All in all, no single technology is available for processing multiple materials. To show the strong material dependence for each technique, we have adapted a schematic from a published paper¹⁰ as below .

(Limited) available elements for each printing technique that can fabricate metal microstructures, as marked in greens in the periodic tables.

Editorial Note: Reprinted with permission from Ref. 10. Copyright (2017) John Wiley and Sons.

For most printing techniques, deposited materials often require subsequent purification process. Although the electron/ion beams induced deposition can process a variety of materials, the printing speed prohibits their practical applications requiring large areas and the impurity is often inevitable. A variety of materials can also be processed with laser-induced forward transfer, but this technique is mainly restricted to print vertical wires and has minimal lateral dimensions of several micrometers, which is approximately three orders of magnitude larger as compared with that demonstrated in our work.

In our work, the strategy for in-situ printing of size-selected aerosol nanoparticles decouples the material dependence on the printing. The particle source can generate aerosol nanoparticles consisting of a wide range of materials⁶ and these nanoparticles are either singly charged or remain neutral³². We also showed that the spatial distribution of electric fields can be controlled based on the theoretical framework of electrostatics. According to that, a pure geometric model³ was used without the need to consider material influence. The configured fields only deliver charged nanoparticles, despite their material differences. These neutral nanoparticles are then prevented to arrive at the printing areas with a double-layer flow pattern (Fig. 1a). The printing principle is then simplified to the motion of an elementary charge in an electrostatic field. After obtaining the field mappings (i.e., spatial distribution) and strength, we can

then put the charged nanoparticles therein for printing.

One may argue that the different materials of the nanoparticles with varied electronegativity can make the charge probability deviate from one to another. This argument is factually true but it only influences the number concentration of charged nanoparticles, thus determining the printing speed. Therefore, the material decoupling effect is still maintained as we only need to adjust the printing time for different materials (Supplementary Table S5). For making this information clearer, we have modified the text in the main manuscript, reading as:

“The reason for this lies in that the nanoscale fields guide the singly charged NPs, regardless of their materials. Electronegativity may have an influence on the charge probability of different materials, but this property only determines the charge probability (i.e., the number concentration of charged ones). To compensate for this difference in material amount, one only needs to adjust the printing time accordingly.”

The particle size selection criteria (equation 1 in the manuscript) formulated by the authors could potentially serve as an important guideline for any future replication / employment of the current technique. More information should be provided regarding on several important details.

(1) How will the NP density within the aerosol flow affect its size selection.

We appreciate the reviewer for realizing the important guideline. As pointed out by the reviewer, the nanoparticle density can influence the size selection, which acts in two aspects:

a) A high nanoparticle density can lead to unwanted particle growth for forming aggregates⁷, which generally carry ill-defined multiple charges³². Such an effect exacerbates the printing, because we lose the control for forming non-uniform structures. Our work carefully designed the particle source to only generate singlet nanoparticles of sub-5-nm⁷ (their size distributions were provided in Supplementary Fig. 2), each of which carried an elementary charge due to the charging mechanism dominated by ion attachment⁵⁰. Subsequently, these singly charged nanoparticles can then be precisely guided by the nanoscale fields for issuing a high precision in printing.

b) A high density of charged nanoparticles can also generate a space charge effect, which reduces the field strength in an uncontrollable manner⁵¹. As a reference, the charge density of the nanoparticles in our system was approximately $1 \times 10^6 \text{ cm}^{-3}$, far below that required for generating the space charge effect^{51,52}. For adding this discussion, we have changed the text to:

“To maintain the guideline feature of Equation (1), we carefully controlled the density of the NPs to prevent particle agglomeration⁵³ (thereby decreasing the likelihood of carrying multiple charges that are ill-defined) and reduce the space charge effect⁵¹ (charged NP density $< 1 \times 10^8 \text{ cm}^{-3}$).”

(2) How are the parameters within the Cunningham correction factor (e.g., α and β) determined, are they material dependent or process dependent?

The parameters in the Cunningham correction factor are all empirical ones⁵⁴ and neither

materials nor process parameters influence their values. In order to eliminate this confusion, we removed all Greek symbols and simply put the numerical values therein, reading as

" According to the definition of electrical mobility:

$$Z_p = \frac{neC}{3\pi\eta D_p};$$

where C is the Cunningham correction factor:

$$C = 1 + \frac{2\lambda}{D_p} \left[1.165 + 0.483 \cdot \exp\left(-\frac{0.997D_p}{2\lambda}\right) \right]; "$$

(3) The validation of the criteria is made with Au NPs (Fig. 1f,g). Did the authors also validate the criteria with other types of NPs to confirm its general applicability?

The reviewer is right that the criterion has general applicability, as we used it to perform size selection of multiple materials (Au, Ag, Pt, Pd, Au–Ag alloy). With the control of the coupling field, the selection of nanoparticles is only based on their electrical mobilities (a physical parameter that is expressed as the velocity of a charged particle in an electric field of unit strength), so the type of material does not have an effect. To verify that, we collected nanoparticles of different materials on a TEM grid with the same conditions as used in the printing. We then statistically analyzed the TEM images for estimating their size distributions. Despite the differences in materials, the size selection was successfully achieved for selecting sub–3–nm particles, as designed with the help of Equation (1). For addressing this concern from the reviewer, we have provided Supplementary Fig. 2 involving multiple materials, as shown below:

Supplementary Fig. 2: Size distribution of NPs measured by the NANO-SMPS and via analyzing their TEM (after size selection) images. a–e, Size distributions of Au (a), Ag (b), Pd (c), Pt (d) and Au–Ag (e) NPs repeated five times (the variations were presented in a shadow region), showing an averaged geometric mean diameter of 5.7 nm (Au), 4.2 nm (Ag), 6.7 nm (Pd), 5.6 nm (Pt) and 5.5 nm (Au–Ag) and an averaged geometric standard deviation of 1.22 (Au), 1.24 (Ag), 1.36 (Pd), 1.29 (Pt) and 1.26 (Au–Ag). f–j, TEM images and size statistics of Au (f), Ag (g), Pd (h), Pt (i) and Au–Ag (j) NPs after size–selection with a proper setting in Equation (1) for only allowing the printing of sub–3–nm particles. The scale bar is fixed to 20 nm.

Is the determination of the characteristic size R (Fig. 2d) made based on 1 material or several materials? This will affect the general applicability of the analytical model.

We totally understand the reviewer’s concern. The physical model is based on the theoretical framework of electrostatics and independent of materials³. The geometrical parameter R represents the spacing between the edges of two neighboring holes (Fig. 2a), and is solely determined by lithographic processing. The parameter h_c characterizes the territory of the local field. All these parameters are derived from a purely geometric model based on the theories in electrostatics field and irrelevant to materials. For testing the general applicability of the model, we switched to different materials but still enable the printing of similar geometries, as shown in Fig. 4i–p. We invite the reviewer to scroll up to the page 19 of this response letter for a check. For clarifying this, we have added the following descriptions in the main manuscript:

“The model is developed based on theoretical framework in electrostatics and is independent of materials. The parameter h_c characterizes the domain governed by the local field and is also related to the feature size. As the motion of singly charged NPs is the same in this field configuration, their materials play no role in printing. The model prediction is verified by the printed Au–Ag nanostructures, but also works for other materials as different materials are controlled to be printed to form similar architectures, as shown in Fig. 4k–p.”

The labels in Fig. S9 are too small to make any clear interpretation.

We have enlarged the layouts for clearly showing the marked values in the images and split the previous figure into multiple ones at different pages (page 9 to 11) in the SI, as shown below:

(continued...)

b

c

(continued...)

Supplementary Fig. 9: Marked geometrical sizes for evaluating the cushion field. a–e, numerically marked cushion heights at a potential of 400 V (a), 500 V (b), 600 V (c),

700 V (d) and 800 V (e) without calibration with the tilt angle. The units for the numerical values marked here are all microns. The scale bar is fixed to 5 μm .

For the demonstration of optical properties made by metal nanoarchitectures in Fig. 5, how does the current process compare to the plasmonic resonance metasurface?

This remark certainly needs more explanations. Metamaterials or metasurfaces are new matter composed of subwavelength structures and provide physical properties that cannot be observed in natural materials. They can flexibly control the electromagnetic response^{55,56}. The properties of metamaterials are mainly determined by their shapes and structural distributions, and the type of material also influences the plasmonic properties^{57,58}. The flexibility to control the geometries by our 3D nanoprinting technique provides an important means to regulate the electromagnetic waves. Traditional fabrication is good at making 2D structures that loses the key feature for having 3D geometries⁵⁹. Our proposed 3D nanoprinting allows the printing of 3D metamaterials and realizes the regulation of their unique electromagnetic responses. Particularly, the multimaterial printing reported in this work enables the possibility to investigate the material-dependent plasmonics (Fig. 5). For clarifying the current process for fabricating metasurfaces/matamaterials, we have added the description as below:

“The flexibility in controlling the architectures and materials with our printing technique provides additional freedom for regulating the electromagnetic responses compared to those only capable of handling few types of materials and geometries. Compelling advantages for adopting this printing technique for fabricating matamaterials/metasures are that the printed 3D nanostructures can be fabricated to be angle sensitive to different light incidences, and their 3D structural characteristics and materials can also be used to tune the plasmonics^{57,58}.”

Additionally, we also mentioned that the metasurfaces can be made to “stand up” for having 3D forms, as we can change their bending angles and intergaps (Fig. 1). The corresponding description reads as:

“Although 2D metasurfaces have been developed, whose bending angles can be varied for manipulating the phase mutation of the scattered light⁶⁰, we fabricated these metasurfaces in 3D forms (Fig. 1i). Such periodic arrangement with different bending angles (Supplementary Fig. 7) and intergaps can be used to control light propagation to a desired direction⁶⁰.”

Have the authors considered any surface protection methods, to prevent potential damages during the actual industrial applications?

We appreciate this comment to help to add practicability for our printing technique. Following the review’s suggestion, we buried the printed nanostructures with photoresist (PR), acting as a protective layer for future industrial applications. Optical images and SEM images confirmed that the printed nanostructures were entirely covered by the PR. These experimental results have been summarized to the newly added Supplementary Fig. 16, as shown below:

Supplementary Fig. 16: Protocols for surface protection. *a*, Process flow for making structural protection. The printed nanostructures were buried by the photoresist to avoid subsequent oxidation and/or mechanical damage. *b*, Optical microscope image and SEM images (*c*) of the structural array after photoresist coverage.

This information has also been added in Discussion section of the main manuscript, reading as:

“For protecting the printed nanostructure arrays for future industrial applications, we successfully cover them with a photoresist layer (Supplementary Fig. S16).”

References used in the response letter

1. Jung, W. *et al.* Three-dimensional nanoprinting via charged aerosol jets. *Nature* **592**, 54–59 (2021).
2. Jung, W. *et al.* 3D Nanoprinting with Charged Aerosol Particles—An Overview. *Accounts Mater. Res.* **2**, 1117–1128 (2021).
3. Jung, Y. H. *et al.* Virtually probing “Faraday three-dimensional nanoprinting”. *Addit. Manuf.* **48**, 102432 (2021).
4. Shin, J. *et al.* Three-dimensional nanoprinting with charged aerosol focusing via an electrified mask. *Addit. Manuf.* **60**, 103206 (2022).
5. Feng, J. *et al.* Green manufacturing of metallic nanoparticles: a facile and universal approach to scaling up. *J. Mater. Chem. A* **4**, 11222–11227 (2016).
6. Feng, J. *et al.* Unconventional alloys confined in nanoparticles: building blocks for new matter. *Matter* **3**, 1646–1663 (2020).
7. Feng, J., Biskos, G. & Schmidt-Ott, A. Toward industrial scale synthesis of ultrapure singlet nanoparticles with controllable sizes in a continuous gas-phase process. *Sci. Rep.* **5**, 15788 (2015).
8. Kohut, A. *et al.* Full range tuning of the composition of Au/Ag binary nanoparticles by spark discharge generation. *Sci. Rep.* **11**, 5117 (2021).
9. Snellman, M., Samuelsson, P., Eriksson, A., Li, Z. & Deppert, K. On-line compositional measurements of AuAg aerosol nanoparticles generated by spark ablation using optical emission spectroscopy. *J. Aerosol Sci.* **165**, 106041 (2022).
10. Hirt, L., Reiser, A., Spolenak, R. & Zambelli, T. Additive manufacturing of metal structures at the micrometer scale. *Adv. Mater.* **29**, 1604211 (2017).
11. Kan, T. & Ajiki, Y. Silicon based mid-infrared photodetectors using plasmonic gold nano-antenna structures. in *2017 19th International Conference on Solid-State Sensors, Actuators and Microsystems (TRANSDUCERS)* 2159–2162 (IEEE, 2017). doi:10.1109/TRANSDUCERS.2017.7994503.
12. Kan, T., Ajiki, Y., Matsumoto, K. & Shimoyama, I. Si process compatible near-infrared photodetector using AU/SI nano-pillar array. in *2016 IEEE 29th International Conference on Micro Electro Mechanical Systems (MEMS)* vols 2016-Febru 624–627 (IEEE, 2016).
13. Li, Z., Butun, S. & Aydin, K. Ultranarrow band absorbers based on surface lattice resonances in nanostructured metal surfaces. *ACS Nano* **8**, 8242–8248 (2014).
14. Ng, R. J. H., Goh, X. M. & Yang, J. K. W. All-metal nanostructured substrates as subtractive color reflectors with near-perfect absorptance. *Opt. Express* **23**, 32597 (2015).
15. Xie, X. *et al.* All-metallic geometric metasurfaces for broadband and high-efficiency wavefront manipulation. *Nanophotonics* **9**, 3209–3215 (2019).
16. Solanki, U. & Mandal, P. All-metal plasmonic metasurface at NIR wavelengths for optical absorption manipulation and refractive index sensing. *Optik (Stuttg.)* **260**, 169107 (2022).
17. Valentine, J. *et al.* Three-dimensional optical metamaterial with a negative

- refractive index. *Nature* **455**, 376–379 (2008).
18. High, A. A. *et al.* Visible-frequency hyperbolic metasurface. *Nature* **522**, 192–196 (2015).
 19. Ebbesen, T. W., Lezec, H. J., Ghaemi, H. F., Thio, T. & Wolff, P. A. Extraordinary optical transmission through sub-wavelength hole arrays. *Nature* **391**, 667–669 (1998).
 20. Liu, H. & Lalanne, P. Microscopic theory of the extraordinary optical transmission. *Nature* **452**, 728–731 (2008).
 21. Neshev, D. N. & Miroshnichenko, A. E. Enabling smart vision with metasurfaces. *Nat. Photonics* **17**, 26–35 (2023).
 22. Fan, J., Montemagno, C. & Gupta, M. 3D printed high transconductance organic electrochemical transistors on flexible substrates. *Org. Electron.* **73**, 122–129 (2019).
 23. Massetti, M. *et al.* Fully 3D-printed organic electrochemical transistors. *npj Flex. Electron.* **7**, 11 (2023).
 24. Guo, X., Xue, Z. & Zhang, Y. Manufacturing of 3D multifunctional microelectronic devices: challenges and opportunities. *NPG Asia Mater.* **11**, 29 (2019).
 25. Karnaushenko, D., Kang, T., Bandari, V. K., Zhu, F. & Schmidt, O. G. 3D Self-Assembled Microelectronic Devices: Concepts, Materials, Applications. *Adv. Mater.* **32**, 1902994 (2020).
 26. Tan, H. W., Choong, Y. Y. C., Kuo, C. N., Low, H. Y. & Chua, C. K. 3D printed electronics: Processes, materials and future trends. *Prog. Mater. Sci.* **127**, 100945 (2022).
 27. Yang, L. *et al.* Laser printed microelectronics. *Nat. Commun.* **14**, 1103 (2023).
 28. Hensleigh, R. *et al.* Charge-programmed three-dimensional printing for multi-material electronic devices. *Nat. Electron.* **3**, 216–224 (2020).
 29. Nassar, H. & Dahiya, R. Fused Deposition Modeling-Based 3D-Printed Electrical Interconnects and Circuits. *Adv. Intell. Syst.* **3**, 2100102 (2021).
 30. Ren, P. & Dong, J. Direct Fabrication of VIA Interconnects by Electrohydrodynamic Printing for Multi-Layer 3D Flexible and Stretchable Electronics. *Adv. Mater. Technol.* **6**, 2100280 (2021).
 31. Han, F. *et al.* Three-dimensional nanofabrication via ultrafast laser patterning and kinetically regulated material assembly. *Science (80-.).* **378**, 1325–1331 (2022).
 32. Hinds, W. C. & Zhu, Y. *Aerosol Technology: Properties, Behavior, and Measurement of Airborne Particles*, 3rd Edition | Wiley. 448 (2022).
 33. Chen, C. *et al.* Macroscale and microscale fracture toughness of microporous sintered Ag for applications in power electronic devices. *Acta Mater.* **129**, 41–51 (2017).
 34. Xia, Y. *et al.* Bulk diffusion regulated nanopore formation during vapor phase dealloying of a Zn-Cu alloy. *Acta Mater.* **238**, 118210 (2022).
 35. Chen, C. & Suganuma, K. Microstructure and mechanical properties of sintered Ag particles with flake and spherical shape from nano to micro size. *Mater.*

- Des.* **162**, 311–321 (2019).
36. Chen, C. *et al.* Mechanical Deformation of Sintered Porous Ag Die Attach at High Temperature and Its Size Effect for Wide-Bandgap Power Device Design. *J. Electron. Mater.* **46**, 1576–1586 (2017).
 37. Zeng, M. *et al.* High-throughput printing of combinatorial materials from aerosols. *Nat. 2023 6177960* **617**, 292–298 (2023).
 38. de Miguel, G., Vicidomini, G., Harke, B. & Diaspro, A. Linewidth and writing resolution. in *Three-Dimensional Microfabrication Using Two-Photon Polymerization* 351–384 (Elsevier, 2020). doi:10.1016/B978-0-12-817827-0.00008-4.
 39. Zhang, P. *et al.* Line width prediction and mechanical properties of 3D printed continuous fiber reinforced polypropylene composites. *Addit. Manuf.* **61**, 103372 (2023).
 40. Li, Q. *et al.* Mechanical nanolattices printed using nanocluster-based photoresists. *Science (80-.)*. **378**, 768–773 (2022).
 41. Faruk, O., Alkadi, F., Kiki, M. & Choi, J. Conformal 3D printing of a polymeric tactile sensor. *Addit. Manuf. Lett.* **2**, 100027 (2022).
 42. Li, D., Hong, H., Sun, X., Zhang, Y. & Liu, Y. Near thermal-electric field controlled electrohydrodynamic 3D printing of high aspect ratio microstructures. (2023).
 43. Onses, M. S., Sutanto, E., Ferreira, P. M., Alleyne, A. G. & Rogers, J. A. Mechanisms, Capabilities, and Applications of High-Resolution Electrohydrodynamic Jet Printing. *Small* **11**, 4237–4266 (2015).
 44. Zhang, B., He, J., Li, X., Xu, F. & Li, D. Micro/nanoscale electrohydrodynamic printing: from 2D to 3D. *Nanoscale* **8**, 15376–15388 (2016).
 45. Zenou, M., Sa’ar, A. & Kotler, Z. Laser jetting of femto-liter metal droplets for high resolution 3D printed structures. *Sci. Rep.* **5**, 17265 (2015).
 46. Maruo, S. & Saeki, T. Femtosecond laser direct writing of metallic microstructures by photoreduction of silver nitrate in a polymer matrix. *Opt. Express* **16**, 1174 (2008).
 47. Galliker, P. *et al.* Direct printing of nanostructures by electrostatic autofocussing of ink nanodroplets. *Nat. Commun.* **3**, 890 (2012).
 48. Ahn, B. Y. *et al.* Omnidirectional printing of flexible, stretchable, and spanning silver microelectrodes. *Science (80-.)*. **323**, 1590–1593 (2009).
 49. Huang, Z., Shao, G. & Li, L. Micro/nano functional devices fabricated by additive manufacturing. *Prog. Mater. Sci.* **131**, 101020 (2023).
 50. Fuchs, N. A. On the stationary charge distribution on aerosol particles in a bipolar ionic atmosphere. *Geofis. Pura e Appl.* **56**, 185–193 (1963).
 51. Alonso, M., Alguacil, F. J. & Kousaka, Y. Space-charge effects in the differential mobility analyzer. *J. Aerosol Sci.* **31**, 233–247 (2000).
 52. Intra, P. & Tippayawong, N. An Overview of Unipolar Charger Developments for Nanoparticle Charging. *Aerosol Air Qual. Res.* **11**, 187–209 (2011).
 53. Feng, J. *et al.* General Approach to the Evolution of Singlet Nanoparticles from

- a Rapidly Quenched Point Source. *J. Phys. Chem. C* **120**, 621–630 (2016).
54. Kim, J. H., Mulholland, G. W., Kukuck, S. R. & Pui, D. Y. H. Slip Correction Measurements of Certified PSL Nanoparticles Using a Nanometer Differential Mobility Analyzer (Nano-DMA) for Knudsen Number From 0.5 to 83. *J. Res. Natl. Inst. Stand. Technol.* **110**, 31–54 (2005).
 55. Smith, D. R., Pendry, J. B. & Wiltshire, M. C. K. Metamaterials and Negative Refractive Index. *Science (80-.)*. **305**, 788–792 (2004).
 56. Shalaev, V. M. Optical negative-index metamaterials. *Nat. Photonics* **1**, 41–48 (2007).
 57. Saleh, B. E. A. & Teich, M. C. *Fundamentals of Photonics*. (John Wiley & Sons, Inc., 1991). doi:10.1002/0471213748.
 58. Maier, S. A. *Plasmonics: Fundamentals and Applications*. (Springer US, 2007). doi:10.1007/0-387-37825-1.
 59. Zhu, Y. *et al.* Recent advancements and applications in 3D printing of functional optics. *Addit. Manuf.* **52**, 102682 (2022).
 60. Yu, N. *et al.* Light propagation with phase discontinuities: Generalized laws of reflection and refraction. *Science (80-.)*. **334**, 333–337 (2011).

REVIEWERS' COMMENTS

Reviewer #1 (Remarks to the Author):

I have read the revised manuscript and the authors have, to a large degree, satisfied my concerns. I would, however, like to point out some minor issues that I feel must be addressed before this manuscript is suitable for publication.

- (i) Line 180 – I suggest changing the wording to “and the results showed no obvious porosity” since any porosity derived from a collection of sub-5 nm particles would not be resolvable on the images shown.
- (ii) The ordering of the supplementary figures in the manuscript goes from Fig 5 to 7 to 6 to 12 to 8. This should be corrected.
- (iii) The figure caption to Figure 2 refers to six “R” values (Lines 237-239) but only 5 are displayed in Figure 2e-i. Some clarification is required.
- (iv) Lines 482-484 refer to the electrode configuration for deriving Au-Ag alloyed nanostructures, but no mention is made of Ni-Ti and Ni-Cr-Co-Mo-Ag alloys. Moreover, it is unclear how the Ni-Cr-Co-Mo-Ag alloyed structures can be fabricated using a two-electrode setup unless an alloyed electrode is used.
- (v) Supplementary Figure 6d and 6e seem to be showing the same thing.
- (vi) It is unclear what is meant by “ones” and “designed downward” in lines 155-157 of Supplementary Information – some rewording is required.
- (vii) Supplementary Figure 15 is never referred to in the manuscript.
- (viii) The SEM scale bar information for the Supplementary Figures 8b-i and Figure 16c are illegible.
- (viii) I would also suggest a careful reading of the manuscript by all authors since there seem to be more than a few sloppy errors in the revised manuscript.

Reviewer #3 (Remarks to the Author):

The authors have sufficiently addressed the questions raised by the current reviewer. He would thus like to recommend acceptance for this manuscript.

We thank the reviewers for their efforts to help us to improve the manuscript. Our response is presented in red font, as shown below.

Reviewer #1 (Remarks to the Author):

I have read the revised manuscript and the authors have, to a large degree, satisfied my concerns. I would, however, like to point out some minor issues that I feel must be addressed before this manuscript is suitable for publication.

We thank the reviewer again for pointing out a number of stylistic corrections. All these issues have been fully addressed as shown below.

(i) Line 180 – I suggest changing the wording to “and the results showed no obvious porosity” since any porosity derived from a collection of sub-5 nm particles would not be resolvable on the images shown.

Following the suggestion from the reviewer, we have reworded the sentence, reading as:

“whose SEM image showed no obvious porosity.”

(ii) The ordering of the supplementary figures in the manuscript goes from Fig 5 to 7 to 6 to 12 to 8. This should be corrected.

Thanks again for this correction. We apologize for this mistake. We are sure that the ordering of the supplementary figures is correctly mentioned in the main manuscript. For easy check, we have highlighted in yellow for the supplementary figures and tables mentioned in the manuscript.

(iii) The figure caption to Figure 2 refers to six “R” values (Lines 237-239) but only 5 are displayed in Figure 2e-i. Some clarification is required.

Figure 2e-i correspond to the five different potentials applied. At each potential, we arranged two SEM images with six rows of the printed nanostructures in total and each arrow represents a value of R . For avoiding the confusion, we have modified the figure caption, reading as:

“e-i) SEM images of 3D nanostructures printed at five different potentials (400, 500, 600, 700 and 800 V). The substrate has an array of opening holes with different values of R (2, 3, 4, 5, 6, and 7 μm), which represents the distance between the edges of neighboring holes and the value thereof is fixed for the same row of printed nanostructures.”

(iv) Lines 482-484 refer to the electrode configuration for deriving Au-Ag alloyed nanostructures, but no mention is made of Ni-Ti and Ni-Cr-Co-Mo-Ag alloys. Moreover, it is unclear how the Ni-Cr-Co-Mo-Ag alloyed structures can be fabricated using a two-electrode setup unless an alloyed electrode is used.

We have added this description as:

“Similarly, Ni-Cr-Co-Mo-Ag nanostructures were printed from the NP source that used a pair of Ni-Cr-Co-Mo and Ag electrodes.”

(v) Supplementary Figure 6d and 6e seem to be showing the same thing.

The reviewer is right. We have removed Figure 6e, and the updated figure now appears

as below:

Supplementary Fig. 7: Nanoscale precision. *a*, SEM image of array nanostructures from Fig. 1j. *b*, An area of multiple line structures corresponding to the SEM image shown in Fig. 1k. The substrate for printing was patterned by an array of channels with a width of 150 nm, and the cracks formed due to sputtering Au layer on the PR surface. *c*, *d*, SEM images for showing sub-20-nm line width. *e*, *f*, Tilt views of the lines are actually true 3D structures, called nanowalls with an aspect ratio of approximately 3–5 (height: 60 nm, as marked in the right panel in *f* with a tilt angle of 53° for imaging, width: 24 nm, length: 10 000 nm). Scale bars: *a*, 1 μm ; *b*, *c*, *e*, *f*, 100 nm; *d*, 40 nm.

(vi) It is unclear what is meant by “ones” and “designed downward” in lines 155-157 of Supplementary Information – some rewording is required.

We have reworded this description, reading as:

“The image consisted 8 square areas in a row and 4 areas in a column and in total, there were 32 areas distributed over the substrate. Each area had the printed nanostructure arrays in different forms. In printing, the gas flow was designed downward along the image, as marked by the arrows.”

For convenience, we have also copied the supplementary figure to here:

Supplementary Fig. 9: Printing metal 3D nanostructures over large areas. Scale bars: b, c, 500 μm ; d, e, f, 10 μm ; g, h, i, 2 μm .

(vii) Supplementary Figure 15 is never referred to in the manuscript.

Indeed, Supplementary Figure 15 was not mentioned in the manuscript. After correcting its ordering mentioned in the manuscript, we have referred to it in the manuscript as: “Hereafter, their interactions are characterized via IR spectroscopy (Supplementary Fig. 13) with a wavelength matching the feature sizes of the 3D-printed structures presented in Fig. 6a and b.”

(viii) The SEM scale bar information for the Supplementary Figures 8b-i and Figure 16c are illegible.

We have increased the readability to the scale bars. One figure was already shown above to the reply of reviewer’s comment (vi) and the other one is shown below:

Supplementary Fig. 16: Protocols for surface protection.

(viii) I would also suggest a careful reading of the manuscript by all authors since there seem to be more than a few sloppy errors in the revised manuscript.

We appreciate the reviewer's careful and through check. All the authors have carefully read the manuscript and confirmed that this version is error free.

Reviewer #3 (Remarks to the Author):

The authors have sufficiently addressed the questions raised by the current reviewer. He would thus like to recommend acceptance for this manuscript.

We thank the reviewer again for raising constructive comments that made our manuscript move to this stage.